# GSG1L-containing AMPA receptor complexes are defined by their spatiotemporal expression, native interactome and allosteric sites

Amanda M. Perozzo [1,2], Jochen Schwenk [3], Aichurok Kamalova[4], Terunaga Nakagawa[4,5], Bernd Fakler [3,6] & Derek Bowie [2] ✉

Transmembrane AMPA receptor regulatory proteins (TARPs) and germ cell-specific gene 1-like protein (GSG1L) are claudin-type AMPA receptor (AMPAR) auxiliary subunits that profoundly regulate glutamatergic synapse strength and plasticity. While AMPAR-TARP complexes have been extensively studied, less is known about GSG1L-containing AMPARs. Here, we show that GSG1L's spatiotemporal expression, native interactome and allosteric sites are distinct. GSG1L generally expresses late during brain development in a region-specific manner, constituting about 5% of all AMPAR complexes in adulthood. While GSG1L can co-assemble with TARPs or cornichons (CNIHs), it also assembles as the sole auxiliary subunit. Unexpectedly, GSG1L acts through two discrete evolutionarily-conserved sites on the agonist-binding domain with a weak allosteric interaction at the TARP/KGK site to slow desensitization, and a stronger interaction at a different site that slows recovery from desensitization. Together, these distinctions help explain GSG1L's evolutionary past and how it fulfills a unique signaling role within glutamatergic synapses.

Claudins were first discovered in chicken liver as being integral membrane proteins necessary for the formation of tight junctions[1]. Since then, an extended family of claudin proteins has been identified that fulfills diverse roles in numerous cell types and organisms from nematodes to humans[2]. In the mammalian brain, claudins have evolved to perform specialized roles at glutamatergic synaptic junctions where they regulate neurochemical transmission mediated by the excitatory neurotransmitter, L-glutamate (L-Glu)[3,4]. Critical claudin proteins found at these synapses include AMPA receptor (AMPAR) auxiliary proteins such as the Type I and II transmembrane AMPAR regulatory proteins (TARPs)[5], as well as germ cell-specific gene 1-like protein (GSG1L), a distant homolog of TARPs[6–8]. TARPs and GSG1L assemble with the pore-forming subunits of AMPARs, which promotes the forward trafficking of receptors to the plasma membrane and modifies both their gating behavior[9,10] and pore properties[11–14].

TARPs and GSG1L are structurally related, both possessing the archetypical claudin membrane topology of cytoplasmic N- and C-termini, four transmembrane domains and two extracellular loops, with the length of the first loop (ExL1) being the primary distinction between them[15,16]. Despite a similar overall topology, TARPs and GSG1L tend to exert opposing positive or negative effects on AMPARs, respectively, in terms of recovery from desensitization[6,7], ion permeation[17], and synaptic transmission[18–20]. These functional distinctions most likely reflect their evolutionary past, with GSG1L emerging later than TARPs as an AMPAR auxiliary subunit[8]. How their marked functional differences can be explained in structural terms, however, has evaded any clear explanation. Although some

[1]Integrated Program in Neuroscience, McGill University, Montreal, QC H3A 1A1, Canada. [2]Department of Pharmacology and Therapeutics, McGill University, Montreal, QC H3G 1Y6, Canada. [3]Institute of Physiology, Faculty of Medicine, University of Freiburg, Hermann-Herder-Str. 7, 79104 Freiburg, Germany. [4]Department of Molecular Physiology and Biophysics, Vanderbilt Brain Institute, Vanderbilt University School of Medicine, Nashville, TN 37232, USA. [5]Center for Structural Biology, Vanderbilt University School of Medicine, Nashville, TN 37232, USA. [6]Signaling Research Centers BIOSS and CIBSS, University of Freiburg, Schänzlestr. 18, 79104 Freiburg, Germany. ✉e-mail: derek.bowie@mcgill.ca

aspects of TARP modulation of AMPAR gating have been tied to the KGK motif, an evolutionarily-conserved regulatory site on the lower D2 lobe of the AMPAR ligand-binding domain (LBD)[21–23], direct experimental evidence for a bona fide regulatory site for GSG1L is lacking.

Proteomic and structural studies of recombinant and native AMPAR complexes have established that TARPs associate with AMPAR tetramers in either a 2:4 or 4:4 arrangement[6,22,24–27] and often partner with cornichons (CNIHs) in hetero-octameric assemblies, especially in the hippocampus[28–31]. In contrast, only two GSG1L subunits have been proposed to bind per tetramer, at least in cryo-EM studies of recombinant protein[15,32]. Based on short-term plasticity observed at corticothalamic synapses, there is an emerging view that individual neurons may establish glutamatergic synapses that are specific for GSG1L[19]; however, the native composition of AMPAR-GSG1L complexes remains to be fully explored.

Here, we present an interdisciplinary and comprehensive study which addresses the spatiotemporal expression profile, native interactome, and functional properties of AMPAR-GSG1L complexes. Unlike TARPs, which are highly expressed across most brain regions throughout development[5,33,34], we find that GSG1L expression is restricted to specific regions and developmental stages in the rodent brain. High-resolution proteomic analyses on native receptor complexes reveal that GSG1L assembles into a unique set of low abundance AMPARs with distinct subunit composition. Contrary to conventional understanding, we show that the main actions of GSG1L are not mediated via the KGK motif, but rather through a separate, evolutionarily-conserved allosteric site. Furthermore, we demonstrate that AMPAR-TARP and AMPAR-GSG1L assemblies rely on subunit coordination between pore-forming and auxiliary subunits, with a dominant role for GluA2, to fine-tune channel gating.

## Results

### GSG1L expression is restricted to specific brain regions and developmental timepoints in the rat brain

Recent work has shown that GSG1L is highly expressed in the anterodorsal (AD) and anteroventral (AV) nuclei of the anterior thalamus (AT) throughout development, where it modulates short-term plasticity[19]. However, a detailed spatiotemporal expression profile of GSG1L remains to be fully characterized. To systematically examine the expression of GSG1L in the brain from P14 to P240, we used a transgenic rat reporting lacZ expression under the control of the endogenous GSG1L promoter[19,35]. Subsequent chromogenic reaction by X-gal produced a dark blue stain, which provides a visual readout of GSG1L promoter activity. The lacZ signal therefore serves as a proxy for GSG1L expression (Fig. 1 and Supplementary Fig. 1).

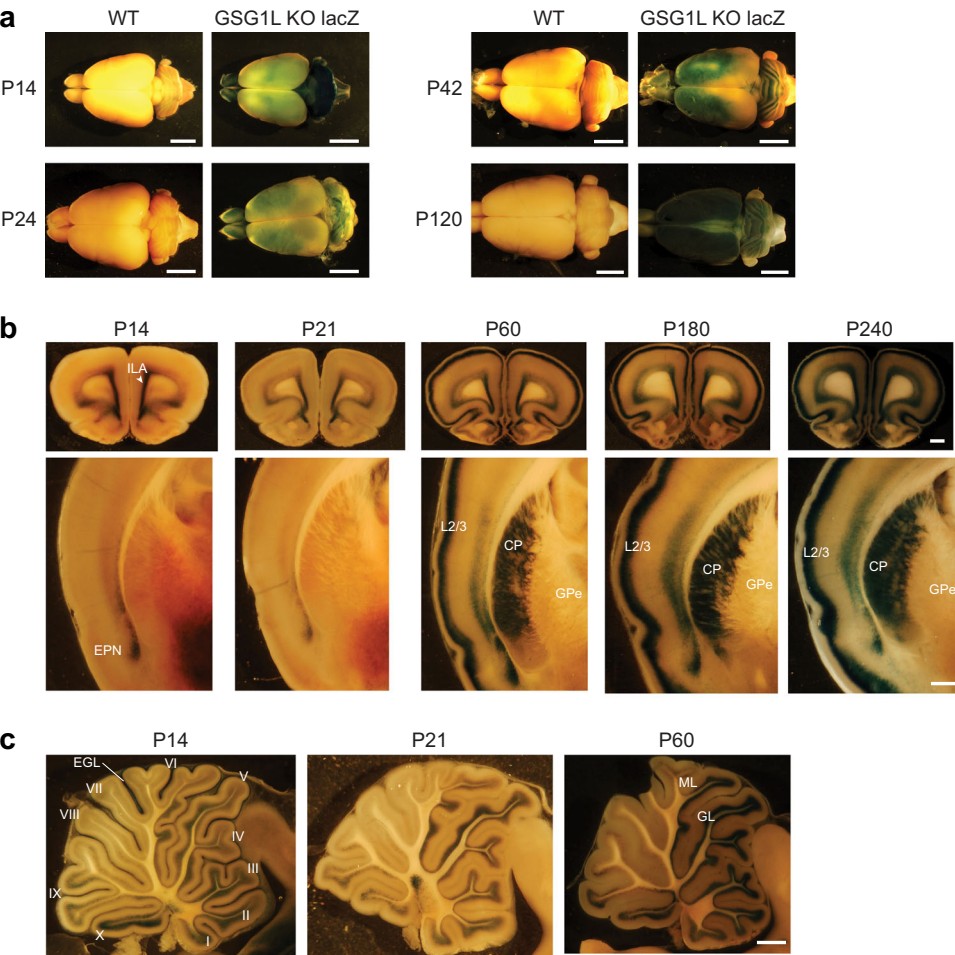

**Fig. 1 | GSG1L has a dynamic, region-specific expression pattern throughout development, with increased expression in the adult rat brain.** a Whole brain lacZ staining of P14, P24, P42, and P120 GSG1L KO and corresponding WT rat brains (scale bar = 5000 μm, n = 2). b GSG1L KO rat brains at P14, P21, P60, P180, and P240 sectioned in the coronal plane at 300 μm and stained for lacZ (scale bar = 1000 μm, n = 2). Labels refer to the infralimbic area (ILA), the endopiriform nucleus (EPN), cortical layer 2/3 (L2/3), the caudate putamen (CP), and the globus pallidus external segment (GPe). c GSG1L KO rat cerebellum at P14, P21, and P60 sectioned in the sagittal plane at 300 μm and stained for lacZ (scale bar = 1000 μm, n = 2). Labels refer to the anterior (I-V) and posterior (VI-X) lobules, external granular layer (EGL), molecular layer (ML), and granular layer (GL).

Intact whole brains of homozygous GSG1L knockout (KO) rats at P14, P24, P42, and P120 were compared with corresponding wildtype (WT) littermates taken as negative controls (Fig. 1a). As expected, no staining was observed in WT controls. By contrast, GSG1L reporter rat brains displayed an evident developmental increase in surface cortical expression of lacZ, with lowest levels at P14 and a continual increase until P120, which was the last timepoint tested. Interestingly, lacZ expression was restricted to defined regions of the cortex, forming a clear boundary of lacZ activity which extended caudally during development (Fig. 1a). Notably, we observed strong lacZ expression in the external granular layer (EGL) of the cerebellum at P14 (Fig. 1a). The EGL contains precursors of granule cells that migrate and differentiate by P20 into the internal granular layer of the cerebellum[36]. Indeed, at later developmental timepoints, we observed high lacZ expression in the granular layer (GL) of the cerebellum (Fig. 1c). Taken together, whole brain staining of the GSG1L KO reporter rat suggests that GSG1L expression levels increase during development, except for limited brain regions, such as the cerebellum, where expression is persistent.

To maximize X-gal penetration and examine GSG1L expression in deeper brain structures, we next generated 300 µm-thick coronal sections from P14, P21, P60, P180, and P240 GSG1L KO brains and stained with X-gal post-sectioning (Fig. 1b and Supplementary Fig. 1). Remarkably, lacZ activity was localized to cortical layer 2/3 (L2/3) neurons, with a clear developmental delay. Staining representative of GSG1L was undetectable in L2/3 at P14 and P21 but was observed at P60 with highest levels at P180-P240, consistent with in situ hybridization data[37]. In the striatum, we found a similar age-dependent increase in lacZ expression with undetectable levels at P14 and P21. At P60, there was strong staining in the caudate putamen (CP) persisting throughout and up to P240. LacZ staining was restricted to the CP with undetectable levels in the globus pallidus (GPe) (Fig. 1b). Our findings reveal that developmental expression of GSG1L is non-uniform across the cortex; while in L2/3 it is restricted to adults, staining in the endopiriform nucleus (EPN) and infralimbic area (ILA) is present throughout postnatal development (Fig. 1b).

Based on the prominent striping pattern observed in the cerebellum of intact whole brains (Fig. 1a, ref. P42), lacZ expression in the cerebellum was further investigated, here using sagittal sections at P14, P21, and P60 (Fig. 1c and Supplementary Fig. 1). We observed distinct lacZ activity in the granular layer (GL) of the cerebellum, while signals were undetectable in the molecular layer (ML) (Fig. 1c). Strikingly, lacZ expression was restricted to certain cerebellar lobules, with distinct segregation to anterior lobes at all three developmental timepoints. Altogether, these observations demonstrate that GSG1L exhibits brain-region, and possibly cell-type-specific, expression during development with global levels increasing throughout maturation.

## Composition of AMPAR-GSG1L complexes in the mature rat brain

Following characterization of the expression profile of GSG1L, we next aimed to identify the precise molecular constituents of native GSG1L-containing AMPAR assemblies. In the mammalian central nervous system (CNS), AMPARs are multiprotein signaling complexes located at extrasynaptic and synaptic sites[38]. The core complex is composed of four pore-forming GluA proteins and up to four auxiliary subunits (TARPs, CNIHs, and/or GSG1L) that bind to two pairs of binding sites[6,24,29]. While it is known that TARP γ8 and CNIH2 co-assemble into the same AMPAR complex, at least in the hippocampus[28,31], the binding partners of GSG1L-containing AMPARs remain largely unexplored. Therefore, we studied the set of interaction partners of GSG1L (interactome; Fig. 2) in the rodent brain using our established multi-epitope affinity purification-mass spectrometry (meAP-MS) approach[6] (see Methods).

Affinity isolation of GSG1L-containing protein complexes from solubilized brain membranes of WT rats (P59) was performed with three different anti-GSG1L antibodies. Source material from GSG1L KO rats served as a stringent negative control. We determined the abundances of all affinity purified proteins and calculated normalized abundance ratios between WT and KO to evaluate the specificity of co-purifications (i.e., target-normalized ratios (tnRs)). Data from all experiments visualized by t-distributed stochastic neighbor embedding (t-SNE) uncovered the proteins that consistently and specifically co-purified with GSG1L by their clustered co-localization with the target (Fig. 2a and Supplementary Fig. 2). Accordingly, GSG1L displayed robust and exclusive interaction with known constituents of the AMPAR interactome[6]: the pore-forming GluA1-4 proteins, a select set of auxiliary subunits (TARPs γ2, γ3, γ8, CNIH2) and three distinct constituents of the receptor periphery (cysteine-knot AMPAR modulating protein 44 (CKAMP44 or Shisa9), proline-rich transmembrane protein 1 (PRRT1), and leucine-rich repeat transmembrane protein 4 (LRRT4)). Noteworthy, none of the AMPAR interactome constituents involved in ER-located biogenesis, including FRRS1L, CPT1c or ABHD6[30,39], were co-purified, suggesting that GSG1L can only bind either to GluA tetramers ready for ER exit or to AMPAR assemblies inserted into the plasma membrane at synaptic/extrasynaptic sites.

For more refined evaluation of the composition of GSG1L-containing AMPARs, the MS-derived abundance values of the main interactors were plotted relative to that of GSG1L (Fig. 2b). GluA2 appeared in an almost 1:1 ratio with GSG1L, while GluA1 and GluA3 were less abundant, pointing towards co-assembly of GSG1L into di- or tri-heteromers as reported in cryo-EM analysis of native AMPARs[29]. Moreover, we estimated the 'average stoichiometry' of GSG1L in AMPARs by determining the abundance ratio of GSG1L and GluA tetramers (sum of all co-purified GluA1-4 proteins divided by four). These results indicated a stoichiometry close to two (1.74 ± 0.31), consistent with structural studies on recombinant GSG1L-bound receptors (e.g.,[32]). The lower abundances determined for TARPs and CNIH2, determined under conditions that are known to preserve stoichiometries[6] (CL-47 buffer), suggest that although GSG1L may form 'conventional' hetero-octameric assemblies with these auxiliary subunits[6,30], it also assembles as the sole inner core constituent in TARP/CNIH-free AMPARs (Fig. 2b).

The relative contribution of GSG1L-containing AMPARs in the rat brain was further analyzed by two-step APs. First, the near-complete pool of GSG1L-containing AMPARs was isolated with anti-GSG1L antibodies, and subsequently, a mixture of anti-GluA1-4 antibodies extracted the entirety of the remaining AMPARs. The quantitative evaluations of MS-analysed AP eluates showed that GSG1L participates in about 5% of all AMPARs in the mature brain (Fig. 2c). Of note, this subpopulation has a particular composition as LRRT4, a known post-synaptic cell adhesion molecule involved in synapse formation[40], showed preferred binding to GSG1L-containing AMPARs (Fig. 2c).

## Members of the claudin superfamily differentially modify AMPAR desensitization

Given the somewhat unique distribution and composition of native AMPAR-GSG1L complexes, we next wanted to further probe the functionality of isolated receptors. To date, the structural basis underlying differential regulation of AMPARs by TARPs (Fig. 3a) and GSG1L (Fig. 3b) is poorly understood. Therefore, we sought to resolve how modulation of AMPAR gating by these claudin proteins can be distinguished from a structure-function perspective. To do so, we first expressed Type I (γ2) and Type II (γ7) TARPs, as well as GSG1L, with homomeric GluA2Q$_{flip}$ AMPARs in HEK293 cells and measured the rate of onset and the extent of desensitization (Fig. 3 and Supplementary Table 1). For comparison, we studied a different class of auxiliary protein, CKAMP44, which also affects AMPAR desensitization[41].

Consistent with previous reports[42–45], the prototypical TARP, γ2, concomitantly slowed entry into and attenuated equilibrium desensitization. Rates into desensitization of GluA2 AMPARs with and without

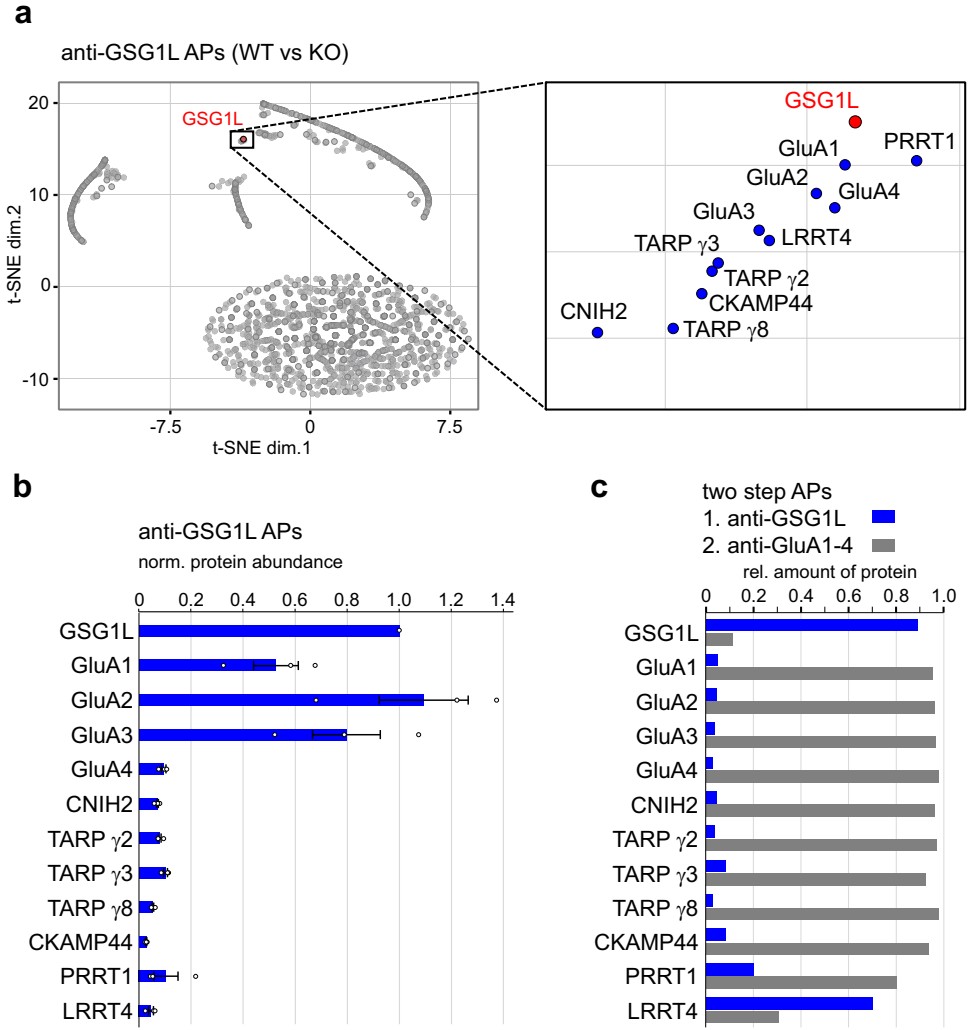

**Fig. 2 | GSG1L is integrated in a subpopulation of native AMPARs with distinct composition. a** t-SNE plot of tnR-values determined for all proteins identified in GSG1L APs with three distinct anti-GSG1L antibodies using target-knockout as negative controls (grey dots; see Methods). The inset on the right highlights the closest interactors, which appear as a cluster with GSG1L. These include all AMPAR pore-forming subunits (GluA1-4) and a set of their proteome components, e.g., TARPs and CNIH2. **b** MS-derived protein abundances determined for the identified GSG1L interactors normalized to GSG1L, data are mean of three experiments ± SEM. **c** Relative amounts of proteins in target-depleting GSG1L APs and subsequent target-depleting anti-GluA1-4 APs. Source data are provided as a Source Data file.

γ2 were best fit as the sum of two exponential functions, with the more prominent fast component corresponding to $8.2 \pm 0.4$ ms (97.7% of response) for GluA2 alone and $12.0 \pm 1.1$ ms (70.7% of response) for GluA2/γ2 (Fig. 3c, d and Supplementary Table 1). Equilibrium desensitization (i.e., steady-state/peak response) significantly increased by 15-fold from $1.4 \pm 0.2\%$ for GluA2 alone ($n = 20$) to $22.2 \pm 1.9\%$ with γ2 ($n = 23$) (Fig. 3e).

Similar to TARP γ2, γ7 and GSG1L both slowed entry into AMPAR desensitization, albeit to a more modest degree (Fig. 3c)[7,21,46–48]. The more robust effect of γ7 was to enhance equilibrium desensitization, resulting in a 4-fold increase relative to GluA2 alone, whereas GSG1L had no appreciable effect (Fig. 3e and Supplementary Table 1). Although the time constants of the fast component of desensitization were similar between GluA2 alone compared to A2 + γ7 and A2 + GSG1L, the contribution of the slow component increased by 3-fold (to 7.1%) and 14-fold (to 33.0%) for γ7 and GSG1L, respectively (Fig. 3d and Supplementary Table 1). TARPs γ2 and γ7, as well as GSG1L, exerted a modest slowing of GluA2 deactivation kinetics (Supplementary Table 1). In contrast, CKAMP44 behaved in an opposite manner to the claudin-related proteins, inducing profound receptor desensitization, as noted previously[49,50]. Co-assembly of GluA2 with CKAMP44

accelerated desensitization kinetics by 1.7-fold to $4.8 \pm 0.2$ ms ($n = 11$) (Fig. 3d). The speeding up of desensitization kinetics was accompanied by a significant 6-fold reduction in the equilibrium response to $0.25 \pm 0.05\%$ of the peak ($n = 11$) (Fig. 3e).

Taken together, these data establish that while all members of the claudin superfamily studied here slow desensitization rates of GluA2 AMPARs, albeit to different degrees, they do not all attenuate equilibrium desensitization (i.e., GSG1L). By contrast, CKAMP44 exerts an opposing modulation on channel gating, enhancing desensitization.

## Both TARPs and GSG1L attenuate AMPAR desensitization through the KGK site

Previous work has shown that mutation of the evolutionarily-conserved KGK motif on the lower lobe of the LBD (Fig. 4a–c) disrupts TARP γ2-dependent slowing of AMPAR entry into desensitization[21]. Whether the KGK motif serves as a common regulatory site for other AMPAR auxiliary proteins, particularly claudins that also attenuate desensitization, has yet to be explored. We therefore compared how mutation of the KGK site (i.e., KGK to single D, herein termed 3D)[21] impacts the gating kinetics of AMPARs in complex with TARPs γ2 and γ7, GSG1L, and CKAMP44 (Fig. 4d–f and Supplementary Table 1).

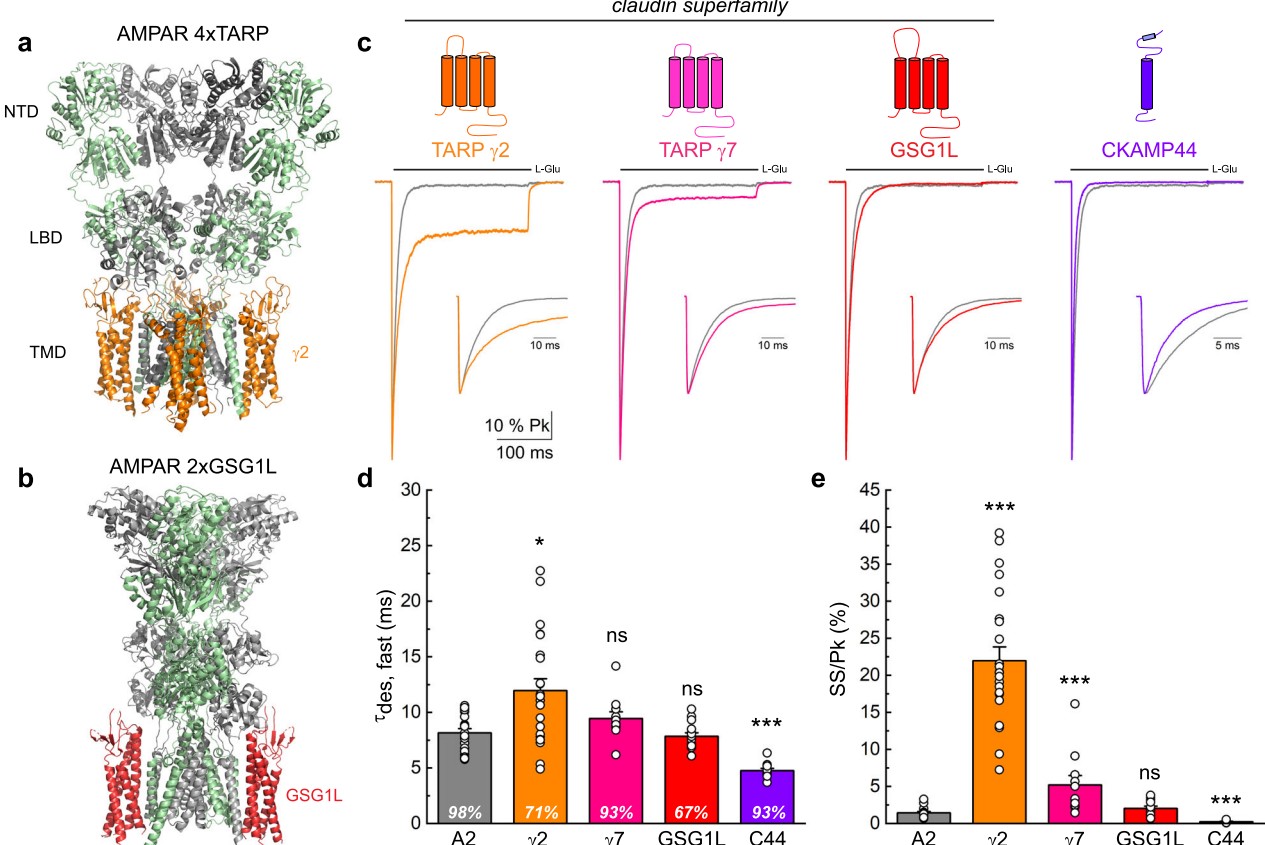

**Fig. 3 | TARPs and GSG1L differentially modify AMPAR desensitization.**
**a**, **b** Cryo-EM structures of the GluA2 tetramer with (**a**) four TARP γ2 subunits (PDB: 5WEO) and (**b**) two GSG1L subunits (PDB: 5VHY). **c** Scaled current responses of GluA2 (patch 170613p1) expressed with γ2 (patch 170630p10), γ7 (patch 180726p3), GSG1L (patch 180423p2) and CKAMP44 (patch 180503p4) upon a 250 ms application of 10 mM L-Glu. **d** Time constants for the fast component of desensitization ($\tau_{\text{des, fast}}$). Percentage on bars indicates the contribution to the overall current decay. **e** Mean equilibrium current amplitude as a percentage of the peak response. For (**d**, **e**) data are mean ± SEM where $n = 20$ for GluA2, $n = 23$ for A2/γ2, $n = 11$ for A2 + γ7, $n = 15$ for A2 + GSG1L and $n = 11$ for A2 + CKAMP44. (ns = not significant, $^*p < 0.05$, $^{***}p < 0.001$, compared to A2 alone, Kruskal-Wallis ANOVA followed by Mann-Whitney $U$ tests with Bonferroni-Holm correction). Source data are provided as a Source Data file.

As expected[21], the 3D mutation significantly attenuated the effect of TARP γ2 on the equilibrium response, resulting in a 4-fold reduction to $5.3 \pm 0.7\%$ ($n = 25$) (Fig. 4d, e) and also disrupted the slowing of desensitization kinetics by γ2 (Fig. 4d, f). Noted previously[21], the 3D mutant itself exhibited faster decay kinetics relative to A2 WT receptors (Fig. 4f and Supplementary Table 1) but indistinguishable steady-state/peak responses (Fig. 4e). For this reason, we also compared the desensitization kinetics of 3D receptors bound by auxiliary subunits with 3D receptors alone (Fig. 4f). In keeping with this, the fast component of desensitization exhibited by 3D/γ2 receptors was $7.9 \pm 0.7$ ms ($n = 25$), remaining similar to 3D receptors in the absence of TARP γ2 ($5.9 \pm 0.2$ ms, $n = 18$) (Fig. 4f), and the slow component was nearly 2-fold faster (Supplementary Table 1). The same general trend was observed for other claudin-related proteins, whereby mutation of the KGK site diminished the effects of both γ7 and GSG1L on entry into desensitization (Fig. 4d, f). For AMPARs co-expressed with γ7, the equilibrium response significantly decreased from $5.2 \pm 1.3\%$ to $0.92 \pm 0.23\%$ ($n = 7$) (Fig. 4e), which was accompanied by an increased contribution of the fast component of desensitization from 92.9% (WT) to 97.4% (3D) ($n = 7$) (Fig. 4f). For AMPARs co-expressed with GSG1L, the contribution of the fast component increased from 67.0% (WT) to 91.0% (3D), leading to an overall acceleration of current decay (Fig. 4f and Supplementary Table 1). More profound desensitization was also manifested by a decrease in the steady-state response from $2.1 \pm 0.3\%$ to $0.75 \pm 0.24\%$ ($n = 11$) (Fig. 4e). The 3D mutant also attenuated the modest slowing of deactivation by the claudin proteins

studied (Supplementary Table 1). In contrast, mutation of the KGK site did not impact the modulatory capacity of CKAMP44, as it continued to induce profound desensitization of 3D receptors, further speeding entry kinetics ($n = 6$) (Fig. 4d, f).

Together, these data demonstrate that the evolutionarily-conserved KGK motif is an allosteric site that is the common target of claudin-related proteins (γ2, γ7, GSG1L), but not CKAMP44. The ability of CKAMP44 to regulate AMPAR gating independent of the KGK motif further reveals that this auxiliary protein must operate through a distinct structural pathway.

## GSG1L slows recovery from desensitization through a separate allosteric site

TARP γ2 has been shown to speed up recovery from AMPAR desensitization (depending on the composition of the pore-forming subunits)[15,38,42,51]. In contrast, GSG1L profoundly slows the recovery process[6,7,15]; as yet, the precise structural basis for this distinction remains unknown. Given that mutation of the KGK site affected both γ2- and GSG1L-mediated slowing of entry into desensitization, we investigated whether this site was also involved in recovery from desensitization (Fig. 5 and Supplementary Table 2). CKAMP44 was also of interest as it has been shown to induce a profound slowing of recovery[41], but whether it does so by the same structural mechanism as GSG1L is unclear.

In agreement with the literature, TARP γ2 modestly sped up GluA2 AMPAR recovery from desensitization ($\tau_{\text{recovery}}$) from $22.3 \pm 2.0$ ms

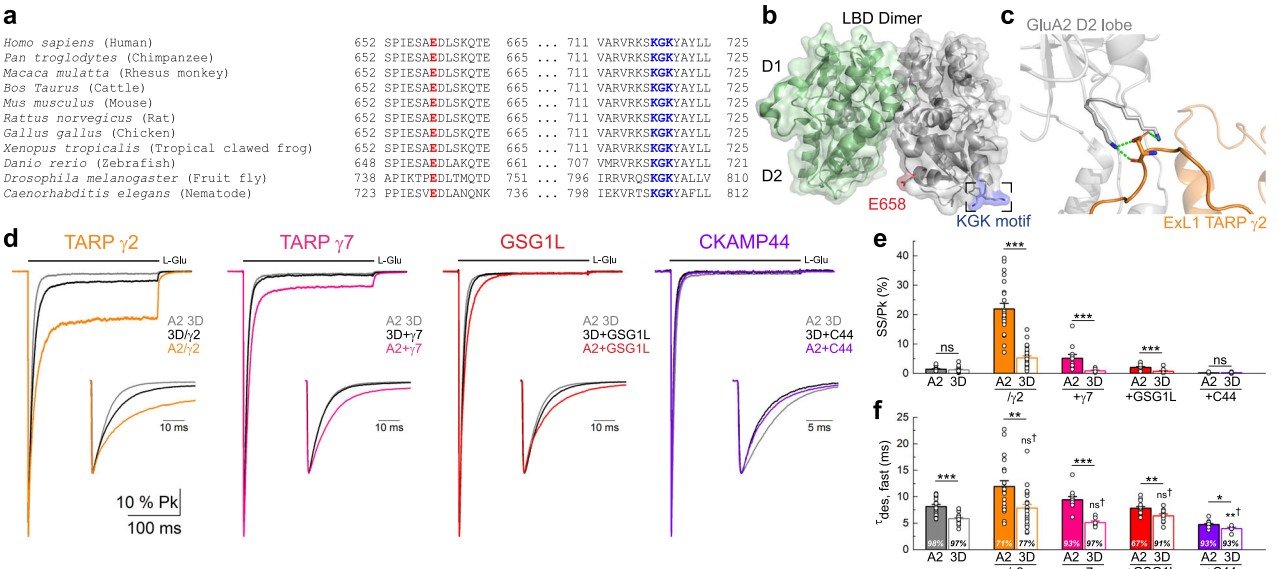

**Fig. 4 | Claudin-related proteins attenuate AMPAR desensitization through the KGK motif. a** Sequence alignment of glutamate receptor subunit 2 gene (*GRIA2*) homologs shows that the E658 (red) and the KGK718-720 (blue) residues are conserved across multiple species. For *D. melanogaster* and *C. elegans*, glutamate receptor IA and non-NMDA ionotropic glutamate receptor subunit GLR-2 genes were used, respectively. Accession numbers for each species: *H. sapiens*, NP_000817.3; *P. troglodytes*, NP_001171923.2; *M. mulatta*, NP_001171942.2; *B. taurus*, NP_001069789.2; *M. musculus*, NP_001077275.1; *R. norvegicus*, NP_058957.1; *G. gallus*, NP_001001775.2; *X. tropicalis*, NP_001135539.1; *D. rerio*, NP_571970.2; *D. melanogaster*, AAF50652.2; *C. elegans*, AAK01094.2. **b** The GluA2 LBD dimer (PBD: 1FTJ) highlighting the E658 residue and KGK motif. **c** Close-up view of the KGK motif on GluA2 and the first extracellular loop (ExL1) on TARP γ2, showing electrostatic interactions between KGK718-720 and D88. Polar contacts are represented by the green dashed lines (PDB: 5KBU). **d** Scaled current responses of GluA2 3D (patch 170623p6) expressed with γ2 (patch 170627p8), γ7 (patch 180410p4), GSG1L (patch 180820p8) and CKAMP44 (patch 180510p1) upon a 250 ms application of 10 mM L-Glu. GluA2 WT (colored trace, previously presented in Fig. 3) is shown for reference. **e** Mean equilibrium current amplitude as a percentage of the peak response. **f** Time constants for the fast component of desensitization ($\tau_{des, fast}$). Percentage on bars indicates the contribution to the overall current decay. For (**e**, **f**), A2 data were first shown in Fig. 3d, e. Data are mean ± SEM where $n = 18$ for GluA2 3D, $n = 25$ for 3D/γ2, $n = 7$ for 3D + γ7, $n = 11$ for 3D + GSG1L and $n = 6$ for 3D + CKAMP44. (ns = not significant, $*p < 0.05$, $**p < 0.01$, $***p < 0.001$, unpaired two-tailed Student's $t$-test or Mann-Whitney $U$ test. ns† = not significant, $**†p < 0.01$, compared to 3D alone, Kruskal-Wallis ANOVA followed by Mann-Whitney $U$ tests with Bonferroni-Holm correction). Source data are provided as a Source Data file.

$(n = 10)$ to $16.7 \pm 0.9$ ms $(n = 6)$, while GSG1L and CKAMP44 had the opposite effect (Fig. 5a, b). The recovery time course with CKAMP44 could also be well fit by a mono-exponential function ($138.4 \pm 19.4$ ms, $n = 8$), whereas GSG1L co-expression resulted in a bi-exponential recovery, with fast and slow time constants of $37.5 \pm 5.4$ ms (16.0%) and $322.5 \pm 34.0$ ms (84.0%), respectively $(n = 9)$ (Fig. 5b and Supplementary Table 2).

When the KGK site was mutated, γ2 modulation of recovery from desensitization was lost $(n = 17)$ (Fig. 5c, d and Supplementary Table 2), suggesting that interactions with the KGK motif are critical for γ2 to regulate both the rates into and out of AMPAR desensitization. Interestingly, mutation of the KGK motif did not disrupt the ability of GSG1L to slow recovery. The recovery time course with GSG1L was bi-exponential, resulting in time constants of $50.5 \pm 12.0$ ms (15.9%) and $300.2 \pm 23.7$ ms (84.1%) $(n = 7)$ (Fig. 5c, d). CKAMP44 was similarly unaffected, slowing recovery of GluA2 3D AMPARs by 7-fold to $148.9 \pm 23.0$ ms $(n = 5)$ (Fig. 5c, d). These observations reveal that γ2 accelerates exit from desensitization through the KGK site, whereas GSG1L and CKAMP44 slow recovery through a different structural pathway.

To evaluate other possible regulatory sites, we focused on the base of the AMPAR LBD as it may be accessible to the ExL1 of GSG1L[15,32]. Based on this, we identified residue Glu658 (E658) which is also evolutionarily-conserved (Fig. 4a) and present across GluA1-4 subunits. This residue was screened by mutagenesis, amongst others, due to its charged nature and location on the lower D2 lobe of the LBD in proximity to the KGK motif (Fig. 4b). Strikingly, we observed that mutation of E658 to Lys (i.e., E658K) eliminated the ability of GSG1L to slow recovery, resulting in a single recovery time constant of $48.0 \pm 4.6$ ms $(n = 8)$ (Fig. 5e, f and Supplementary Table 2).

Importantly, GSG1L continued to slow desensitization and deactivation kinetics of E658K AMPARs, similar to WT receptors, indicating that the mutation did not disrupt GSG1L assembly (Supplementary Fig. 3 and Supplementary Table 1). TARP γ2 continued to accelerate recovery from desensitization of E658K receptors from $47.0 \pm 3.3$ ms $(n = 6)$ to $32.0 \pm 2.6$ ms $(n = 7)$ (Fig. 5e, f), demonstrating that the E658 residue fulfills a role that is distinct from the KGK site. CKAMP44 regulation was also unaffected by the E658K mutation ($180.1 \pm 18.0$ ms, $n = 8$) (Fig. 5e, f), which signifies that GSG1L and CKAMP44 must slow recovery from desensitization through separate allosteric sites. Together, these findings uncover that GSG1L, unlike TARPs, regulates the onset and recovery from AMPAR desensitization through two distinct allosteric sites on the AMPAR LBD. Furthermore, the regulatory effect of the E658 residue is specific to slow recovery/exit from the desensitized state imparted by GSG1L.

## TARPs also modify GluA1/A2 AMPAR heteromer gating via the KGK motif

Since the majority of native AMPARs are heteromers composed of GluA1/A2 subunits[52], we next used our regulatory sites found in GluA2 homomers (Figs. 4 and 5) to better understand how TARPs (Fig. 6) and GSG1L (Fig. 7) modulate GluA2(R)-containing heteromeric gating. Heteromerization was confirmed in each recording by testing for the loss of cytoplasmic polyamine block (Supplementary Fig. 4 and Methods), as we have performed previously[24]. To date, most studies have focused on TARPs where TARP γ2 (or γ8) is tethered to either GluA1 or GluA2, or both, revealing that the degree of TARP modulation of AMPARs is greater with 4 TARP subunits than with 2 subunits[24,26,53,54]. Given this, we tethered TARP γ2 to both GluA1 and GluA2(R) (see Methods) to maximize the TARP phenotype, which allowed us to

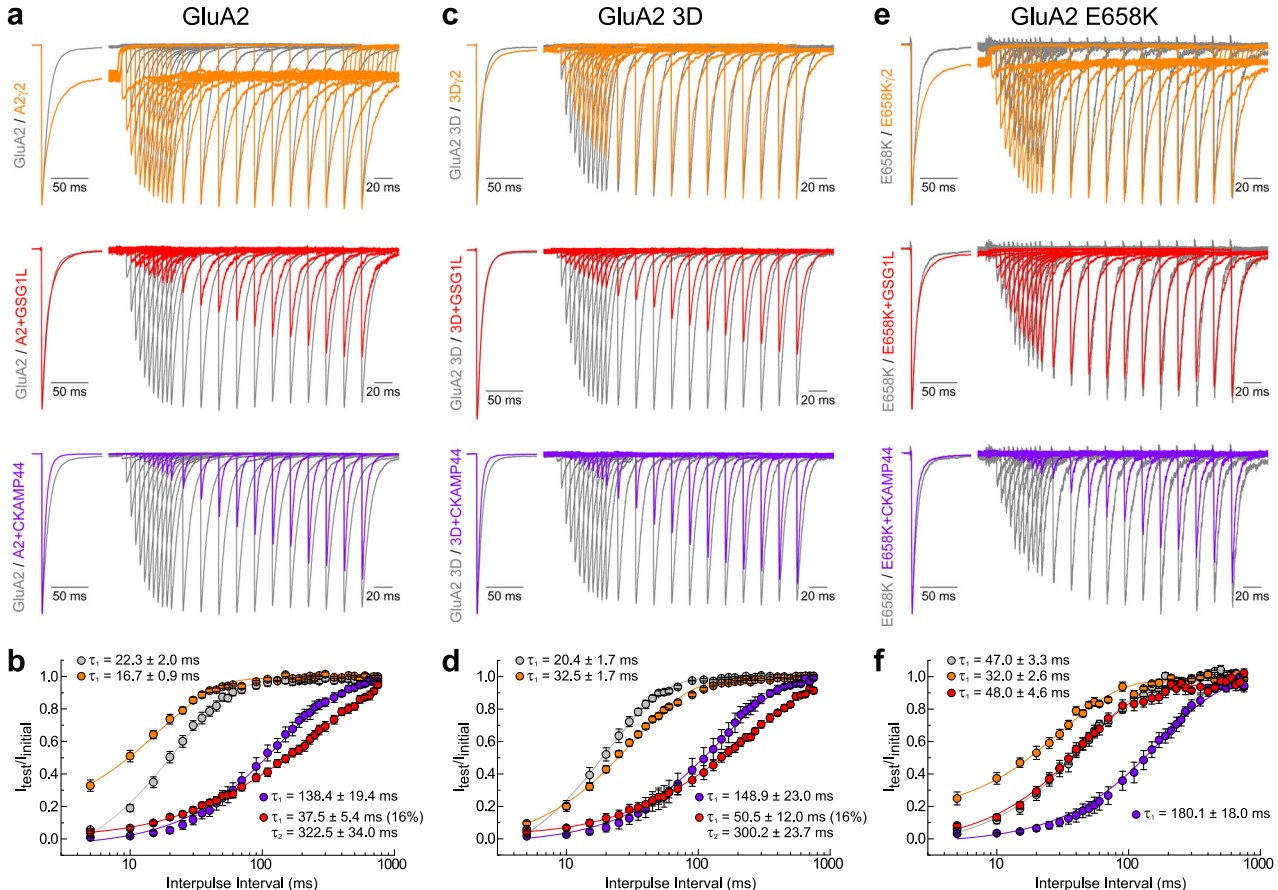

**Fig. 5 | GSG1L slows recovery from desensitization through a unique regulatory site. a** Recovery from desensitization for GluA2 WT receptors (patch 170215p1) expressed with γ2 (patch 170630p10), GSG1L (patch 180820p14) and CKAMP44 (patch 180507p3). **b** Time constants of recovery from desensitization (τ_recovery) for GluA2 WT receptors expressed with auxiliary proteins. The solid lines represent average fits. Data are mean ± SEM where $n = 10$ for GluA2, $n = 6$ for A2/γ2, $n = 9$ for A2 + GSG1L and $n = 8$ for A2 + CKAMP44. **c** Recovery from desensitization for GluA2 3D mutant receptors (patch 170623p6) expressed with γ2 (patch 170131p4), GSG1L (patch 180820p8) and CKAMP44 (patch 180510p9). **d** Same as (**b**) but with 3D mutant receptors. Data are mean ± SEM where $n = 6$ for GluA2 3D, $n = 17$ for 3D/γ2, $n = 7$ for 3D + GSG1L and $n = 5$ for 3D + CKAMP44. **e** Recovery from desensitization for GluA2 E658K mutant receptors (patch 180626p9) expressed with γ2 (patch 180514p8), GSG1L (patch 180723p3) and CKAMP44 (patch 180626p8). **f** Same as (**b, d**), but with E658K mutant receptors. Data are mean ± SEM where $n = 6$ for GluA2 E658K, $n = 7$ for EK/γ2, $n = 8$ for EK + GSG1L and $n = 8$ for EK + CKAMP44. Source data are provided as a Source Data file.

interrogate the individual contributions of GluA1 and GluA2 to channel gating by way of sequentially mutating the KGK site in each subunit (Fig. 6a, cartoon and Supplementary Table 3).

Fully-TARPed heteromers (A1/γ2 + A2/γ2) exhibited a pronounced equilibrium response of 24.0 ± 2.1% and a fast time constant of desensitization of 7.7 ± 0.7 ms ($n = 6$), which were in good agreement with the responses evoked by A1/A2 heteromers freely expressed with γ2 ($n = 7$) (Fig. 6b, c and Supplementary Table 3). These observations reaffirm previous work[21,24,54] that tethering AMPAR subunits to TARPs does not affect the normal functional characteristics of the receptor. Surprisingly, mutation of the KGK motif on GluA1 had no effect on TARP modulation of either the steady-state response or desensitization kinetics ($n = 10$) (Fig. 6b, c and Supplementary Table 3). Given that the 3D mutation in GluA1 homomers disrupted TARP modulation (Supplementary Fig. 5 and Supplementary Table 3), this finding may be explained by a functional dominance of the GluA2 subunit. In keeping with this, subsequent mutation of the KGK motif on GluA2 elicited a significant decrease in the steady-state response by nearly 2-fold to 13.5 ± 1.1% and a concomitant speeding up of desensitization kinetics to 4.9 ± 0.3 ms ($n = 9$) (Fig. 6b, c). Interestingly, mutation of KGK on both GluA1 and GluA2 led to a further reduction in the steady-state response to 5.6 ± 0.6% ($n = 8$), revealing a synergy that occurred only when all pore-forming

subunits were mutated (Fig. 6b). Desensitization kinetics, however, did not speed up beyond mutating GluA2, suggesting that the functional interaction at GluA2 is sufficient to mediate TARP effects on gating kinetics (Fig. 6c and Supplementary Table 3).

Taken together, two important conclusions emerge from these data. Firstly, our observations underscore a dominant role for the GluA2 subunit in TARP modulation of AMPAR heteromer gating, in line with recent structural and functional studies[24,26,31]. As alluded to above, our results cannot be explained by an insensitivity of the GluA1 subunit to TARP modulation via the KGK site, since the 3D mutation in GluA1 homomers abolishes γ2 modulation (Supplementary Fig. 5). Accordingly, our data suggest a functional asymmetry in the architecture of the AMPAR heteromer that determines how TARPs affect the GluA1 and GluA2 subunits during channel gating. Secondly, our data reveal that all four KGK sites need to be mutated to achieve the greatest loss of TARP modulation, demonstrating coordination between subunits within the AMPAR-TARP complex during activation.

## Slow recovery from desensitization is coordinated by GSG1L via GluA2 in AMPAR heteromers
We next examined the structural basis underlying GSG1L-mediated slow recovery from desensitization of GluA1/A2 heteromers (Fig. 7a₁-d₁ and Supplementary Table 4). To do so, we employed a similar

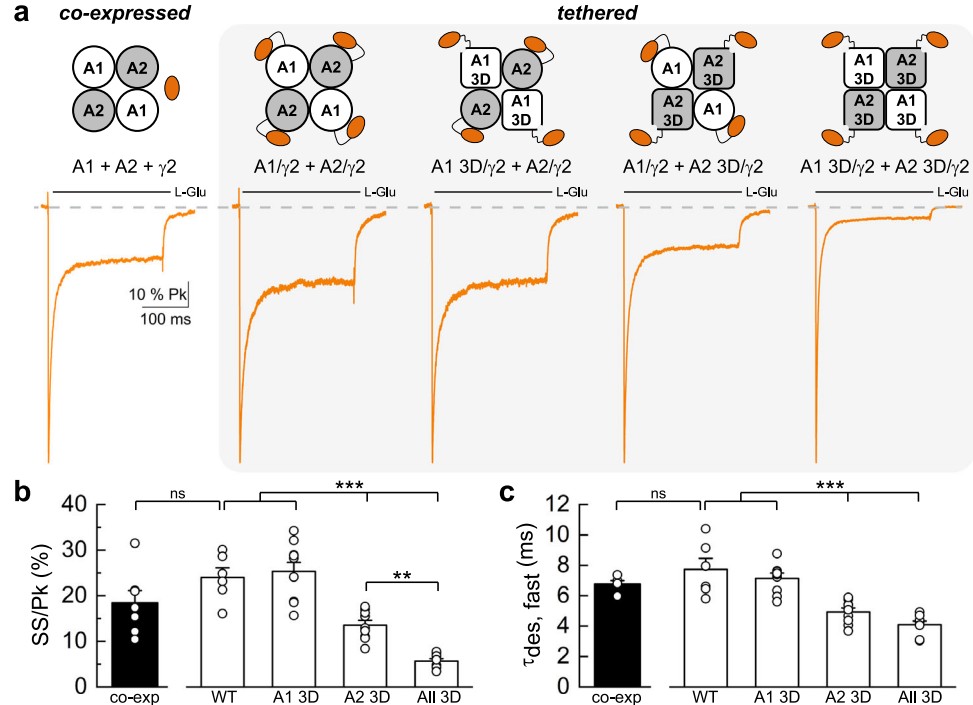

**Fig. 6 | GluA2 dominates gating in heteromeric AMPAR-TARP complexes, which is mediated through the KGK site. a** Scaled current responses of A1/A2 heteromers co-expressed with or tethered to four TARP γ2 in response to a 250 ms application of 10 mM L-Glu. From left to right: co-expressed (co-exp, patch 180821p3), WT tethered heteromer (WT, patch 190822p1), 3D mutation on GluA1 (A1 3D, patch 190822p7), 3D mutation on GluA2 (A2 3D, patch 190826p3) and 3D mutation on all pore-forming subunits (All 3D, patch 190826p9). **b** Mean

equilibrium current amplitude as a percentage of the peak response. **c** Time constants for the fast component of desensitization ($\tau_{des,\ fast}$). For (**b**–**c**), data are mean ± SEM where $n = 7$ for co-exp, $n = 6$ for WT, $n = 10$ for A1 3D, $n = 9$ for A2 3D and $n = 8$ for All 3D. (ns = not significant, unpaired two-tailed Student's $t$-test co-exp vs. WT; **$p < 0.01$, ***$p < 0.001$, one-way ANOVA with Tukey's HSD test). Only significant results are indicated for clarity for tethered receptors. Source data are provided as a Source Data file.

mutagenesis strategy as described in Fig. 6, this time with the E-to-K mutation (EK; residue 651 in GluA1, 658 in GluA2) (ref. Figs. 4 and 5). Important to note, GSG1L was freely expressed with GluA1 and GluA2 subunits in these experiments, since tethering GSG1L disrupted its canonical modulatory properties (Supplementary Figs. 6–8 and Supplementary Tables 3-5). Thus, the responsiveness of receptors to the partial agonist kainate (KA) was also assessed to confirm GSG1L incorporation into AMPAR complexes, as AMPAR heteromers alone are relatively insensitive to KA (Fig. 7a$_2$-d$_2$ and Supplementary Table 5).

As with GluA2 homomers, co-assembly of GSG1L with GluA1/A2 heteromers resulted in a slow, bi-exponential recovery time course with fast and slow time constants of 23.7 ± 3.8 ms (20.1%) and 547.6 ± 62.7 ms (79.9%), respectively ($n = 6$) (Fig. 7a$_1$). When EK was mutated on GluA1 only, GSG1L-bound heteromers still exhibited a slow recovery time course but it was best fit by a mono-exponential function, with a $\tau_{recovery}$ of 338.3 ± 36.7 ms ($n = 7$) (Fig. 7b$_1$). Although this recovery did not exactly match the bi-exponential time course of WT heteromers co-expressed with GSG1L (blue fit line), it was still relatively slow compared to WT and EK mutant heteromers alone (Supplementary Table 4). This distinction may hint at some functional contribution by GluA1 masked by the dominance of GluA2. Indeed, subsequent mutation of the EK site on GluA2 lead to a greater attenuation of GSG1L's effect on the recovery time course, with fast and slow time constants of 80.6 ± 10.7 ms (54.7%) and 446.0 ± 20.5 ms (45.3%), respectively ($n = 7$) (Fig. 7c$_1$). Of note, the contribution of the slow component to the recovery time course was much smaller relative to WT complexes (Supplementary Table 4). Notably, mutation of EK on both GluA1 and GluA2 subunits nearly eliminated slow recovery mediated by GSG1L, resulting in a single $\tau_{recovery}$ of 129.8 ± 8.0 ms ($n = 9$) (Fig. 7d$_1$). Altogether, these data demonstrate a dominant role for GluA2 in dictating the time course of recovery from desensitization of AMPAR-GSG1L heteromers.

## Discussion

The present study provides a comprehensive characterization of the AMPAR-GSG1L complex, advancing our understanding of GSG1L in several important ways. First, we demonstrate that GSG1L expression in the rodent brain is region-specific and developmentally regulated, with striking patterns observed in the cerebellum, caudate putamen, and cerebral cortex. Second, we also shed light on the native GSG1L interactome and find that GSG1L binds to all GluA subunits, with an average stoichiometry of two GSG1L proteins per GluA tetramer. Within native AMPAR complexes, GSG1L can either constitute the inner core alone or together with TARP/CNIH subunits. Finally, we provide structural insights into how GSG1L and TARP γ2 differentially modify AMPAR desensitization in homomeric and heteromeric assemblies. We show that slow recovery by GSG1L is not mediated through the KGK site, but instead regulated by a separate, evolutionarily-conserved allosteric site unique to GSG1L. We reveal that coordination between pore-forming subunits and asymmetry underlie AMPAR-claudin gating, with a privileged role for GluA2 in dictating both TARP and GSG1L modulation. As summarized in Fig. 8 and discussed below, together, these features allow AMPAR-GSG1L signaling complexes to fulfill specialized roles at select glutamatergic synapses.

The first theme that emerges from our work is the unique expression pattern and interactome of native AMPAR-GSG1L complexes. The precise role of GSG1L in the brain represents a relatively new area of study that demands a spatiotemporal mapping of its expression and better understanding of its assembly. For example, initial studies focused on the hippocampus[18,20] where GSG1L promoter activity was recently reported to be low early in development and localized to dentate granule cells (GCs) and CA3 later in development[19] (Supplementary Fig. 1), consistent with in situ hybridization data[37]. Moreover, evidence from the anterior thalamus (AT) suggests that

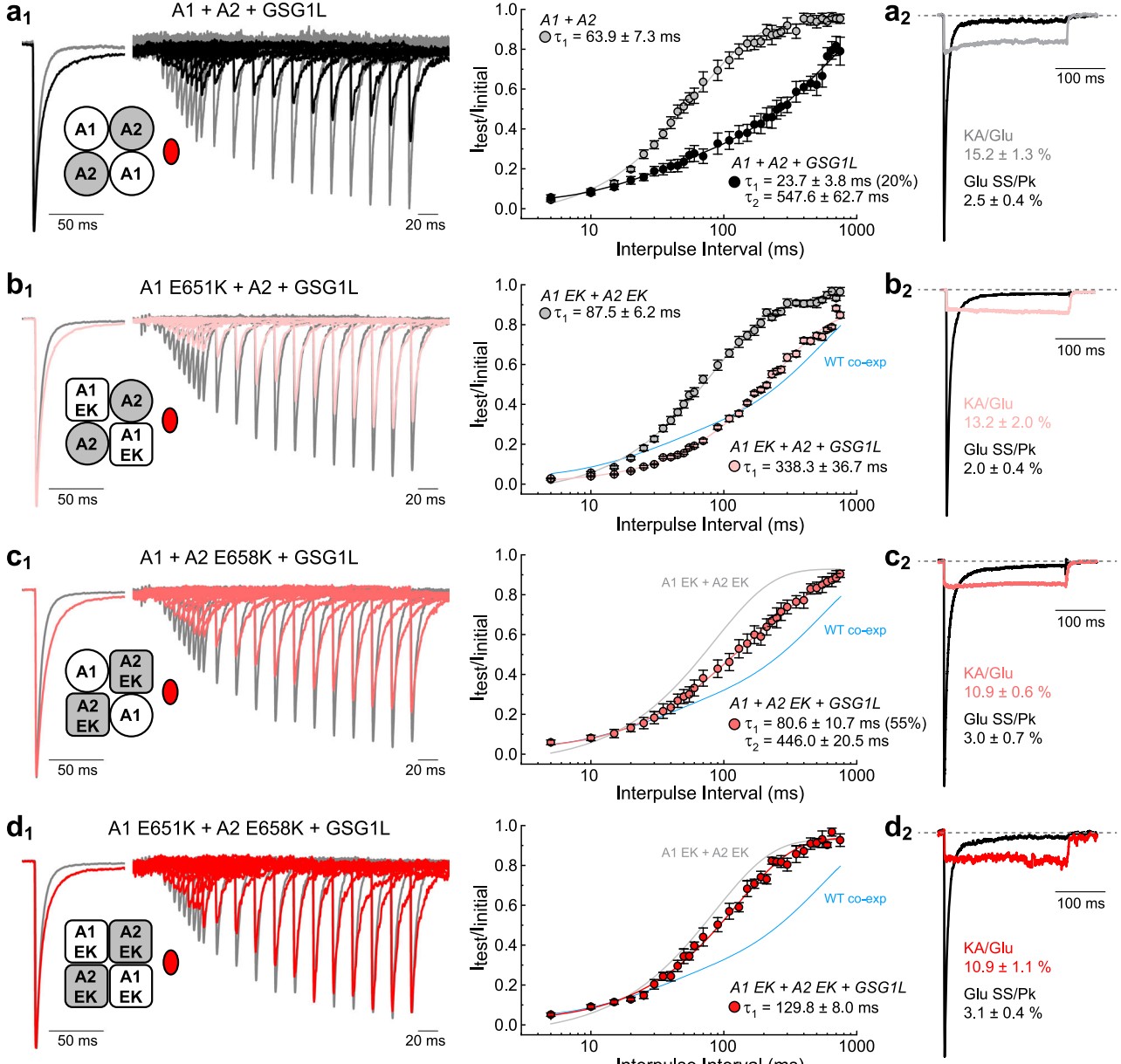

**Fig. 7 | Heteromeric AMPAR-GSG1L complexes exhibit slow recovery from desensitization that is mediated through the EK site. $a_1$** Recovery from desensitization for A1/A2 WT heteromers alone (grey, patch 200714p5) compared to WT receptors co-expressed with GSG1L (black, patch 191104p4). **$b_1$** Recovery from desensitization for mutant GluA1 E651K/GluA2 E658K (i.e., A1 EK/A2 EK) heteromers alone (grey, patch 210504p13) compared to receptors where EK is mutated on GluA1 only and co-expressed with GSG1L (light pink, patch 210429p1). **$c_1$** Same as (**$b_1$**), but compared to receptors where EK is mutated on GluA2 only and co-expressed with GSG1L (dark pink, patch 210422p5). **$d_1$** Same as (**$b_1$**, **$c_1$**), but compared to receptors where EK is mutated on all GluA pore-forming subunits and co-expressed with GSG1L (red, patch 210426p2). For (**$a_1$**–**$d_1$**), the scatter plot depicts the recovery time course and time constants ($\tau_{recovery}$) for each receptor

combination. The solid lines represent average fits of the data as indicated. Data are mean ± SEM where $n = 6$ for A1/A2 WT alone, $n = 6$ for A1/A2 WT with GSG1L, $n = 9$ for A1 EK/A2 EK alone, $n = 7$ for EK on A1 with GSG1L, $n = 7$ for EK on A2 with GSG1L, and $n = 9$ for EK on both A1 and A2 with GSG1L. **$a_2$–$d_2$** Scaled current responses of WT and mutant heteromers co-expressed with GSG1L evoked by 250 ms applications of 10 mM L-Glu or 1 mM KA. A robust KA current confirms incorporation of GSG1L into the receptor complex. The steady-state current (in KA and L-Glu) as a percentage of the peak response in L-Glu is indicated (mean ± SEM). Patch and $n$ numbers: (**$a_2$**) WT, patch 200716p6, $n = 6$; (**$b_2$**) EK on A1, patch 210429p1, $n = 8$; (**$c_2$**) EK on A2, patch 210423p4, $n = 6$; and (**$d_2$**) EK on both A1 and A2, patch 210513p9, $n = 7$. Source data are provided as a Source Data file.

some neurons establish synapses specific for the AMPAR-GSG1L complex[19]. In keeping with this emerging view, our lacZ reporter staining for GSG1L promoter activity shows a high degree of specificity, i.e., expression is not ubiquitous (Fig. 1), and our proteomic data uncover that GSG1L assembles into AMPAR complexes with distinct composition (Fig. 2). Although the signals of the lacZ reporter in the AT are consistent with the functional expression of GSG1L[19] and many signals presented in the current work agree with the Allen Brain Atlas

of the mouse brain[37], caution is still needed because GSG1L promoter activity may not be faithfully reproduced as a result of insertion of the lacZ reporter in the genome. We suggest that endogenous GSG1L expression is most likely found in regions where our lacZ results and the Allen Brain Atlas agree. The excess of GSG1L in affinity purifications in relation to other auxiliary subunits (Fig. 2b) is suggestive for the surprising, preferred occurrence of GSG1L in TARP/CNIH-free AMPARs. As a note of caution, these complexes might be

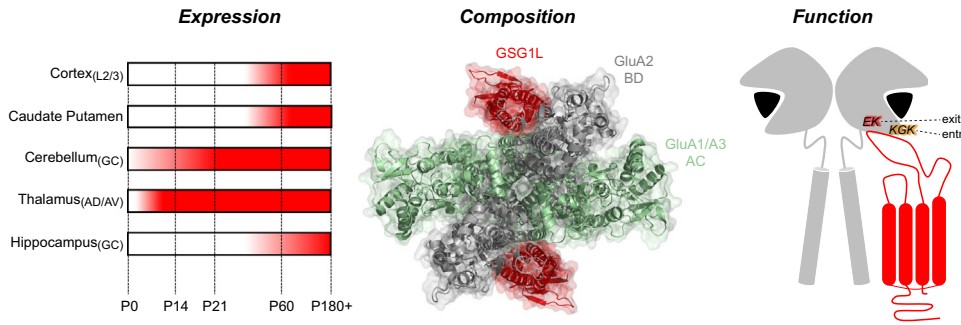

**Fig. 8 | The hallmarks of GSG1L-containing AMPAR complexes.** GSG1L is uniquely identified by three defining features: First, it exhibits distinct temporal expression across different brain regions and cell layers (left). Second, it interacts with GluA2-containing AMPARs at a 2:4 stoichiometry and can assemble in the absence of other auxiliary subunits (middle, PDB: 7RYZ). Third, it slows entry into and recovery/exit out from desensitization through separate allosteric sites located on the lower lobe of the AMPAR LBD (right). GSG1L's actions via the KGK motif can be thought of as secondary compared to its profound regulation of recovery mediated through the EK site.

overrepresented in our experiments, as the binding of GSG1L antibodies could partially destabilize the interaction of other auxiliaries. The apparent solitary role of GSG1L positions it as a unique regulator of AMPAR-mediated neurotransmission. Indeed, functional segregation of AMPAR-auxiliary complexes has been demonstrated at synapses of AD/AV neurons in the AT. TARP γ2 and GSG1L are both expressed in these neurons but exhibit input-specific regulation and likely do not co-assemble[19].

Our study reports that although GSG1L is incorporated into only a subset (about 5%) of all AMPARs in the mature rat brain (Fig. 2c), its expression globally increases throughout development (Fig. 1). One caveat of our proteomic work is that it does not fully capture the dynamics of the GSG1L interactome during development; as such, proteins that assemble at early timepoints or in a developmentally-dependent, cell-type-specific manner may have been overlooked. Nonetheless, in the cortex and caudate putamen (CP), GSG1L exhibits a clear, late developmental onset between P21 and P60, whereas it is expressed earlier in migrating GCs in the cerebellum (Fig. 1). Of note, the CP receives excitatory inputs from the cortex suggesting that GSG1L may be an important constituent of a neuronal circuit widely thought to be involved in motivated behavior[55]. Why GSG1L expression is activated at later developmental stages in these regions is unclear, though CKAMP44, which similarly slows recovery from desensitization, is reportedly expressed throughout the cortex early in development[41]. In both the cortex and cerebellum, GSG1L expression is restricted, namely to layer 2/3 and GCs, respectively. Curiously, expression in the cerebellar GL is not uniform across lobules, with GSG1L localizing primarily to the anterior portion which has been linked to sensorimotor function[56]. Although obvious motor deficits were not observed in GSG1L KO mice[19], it does not necessarily mean that its loss does not impact cerebellar-mediated behaviors, such as eyeblink conditioning[57], as shown for GluA4 KO mice[58]. An intriguing possibility, however, is that GSG1L may be involved in encoding synaptic information in cerebellar GCs, explaining how short-term depression of AMPAR responses can paradoxically affect neuronal gain control[59–61].

Interesting to note, the cerebellar GC is abundant in claudin-type AMPAR auxiliary proteins (i.e., TARPs γ2, γ7, and GSG1L) as well as CKAMP39[49,62], with GluA4 as the major AMPAR subunit[63,64]. Although there is a low widespread abundance of GluA4 based on our proteomic analyses (GSG1L APs; Fig. 2b), it may, however, correspond to most AMPAR complexes expressed by cerebellar GCs. Our findings also indicate that a prominent constituent of GSG1L-containing AMPARs is the transsynaptic adhesion protein, LRRT4, perhaps implying that this protein plays a role in directing the formation and organization of GSG1L-rich synapses. Altogether, this work presents a valuable foundation for future investigation into the impact of GSG1L on AMPAR response fidelity, excitability, and circuit behavior.

Another important theme stemming from our study is that the lower D2 lobe of the AMPA receptor LBD acts as an allosteric hub for GSG1L and TARPs. Our data uncover that claudin homologs, TARPs and GSG1L, target AMPARs at two functionally distinct sites that regulate entry into and exit out of receptor desensitization. Previous studies have already shown that TARP γ2 regulates the onset of desensitization of GluA2 homomers through the KGK site[21]. The present study extends this finding to both Type I and Type II TARPs, as well as GSG1L, revealing that all these claudins slow the onset of desensitization through the same structural mechanism, albeit to varying degrees (Figs. 3 and 4). The importance of the KGK motif in claudin regulation of AMPAR gating is also extended to GluA1 homomers (Supplementary Fig. 5) and GluA1/A2 heteromers (Fig. 6).

In contrast, γ2 and GSG1L have differential effects on recovery from desensitization (Fig. 5). While γ2 modestly speeds up the recovery process via the KGK site, our data show that the profound slowing of recovery by GSG1L is mediated by a single residue that is conserved amongst all AMPAR subunits. Since TARPs and GSG1L share a common ancestry[8], it is tempting to speculate that GSG1L's much weaker effect on the KGK site represents a vestige of its evolutionary past rather than serving any functional purpose. Structural and functional studies have argued that the first extracellular loop of GSG1L, which is comprised of variable sized flexible loops between four β-strands, interacts with AMPARs to slow recovery[15]. In particular, the β1-β2 loop contains 49 amino acids that differ between GSG1L and TARPs and, although not fully resolved in cryo-EM structures, these residues are thought to directly interact with AMPARs. The length of this loop is proposed to underlie the functional interactions that distinguish GSG1L and TARPs, with our work now revealing that the different loops may contact distinct regions on the AMPAR LBD (Fig. 8). Furthermore, drastic structural differences in the symmetrical organization of desensitized-state LBD dimers between AMPAR-TARP and AMPAR-GSG1L complexes have been observed; the loss of two-fold symmetry reported for GSG1L may impact its interactions with the LBD[32]. Taken together, these findings reveal unexpectedly that two important CNS claudin family members, TARP γ2 and GSG1L, have evolved to regulate AMPARs through structurally-distinct and allosterically-separate mechanisms.

In addition to the interface between the extracellular loops and the LBD, functional studies have also demonstrated the importance of AMPAR-TARP interactions at the level of the transmembrane alpha helices and C-termini[65,66], TMD-LBD linkers[67], and even the NTD[68]. In contrast to TARPs and GSG1L, the structural architecture of the AMPAR-CNIH complex indicates that CNIHs are embedded in the plasma membrane, such that interactions occur exclusively at the TMD[69]. To date, the structure of an AMPAR associated with CKAMPs has yet to be resolved. Although CKAMP44 slows recovery from

desensitization, our data demonstrate that the mechanism of allosteric modulation is different from GSG1L.

The final theme relates to the asymmetrical contribution of pore-forming subunits in AMPAR heteromers that is unveiled by claudin-related proteins. Early structural studies revealed that the placement of core subunits within the AMPAR tetramer is critical for channel activation[70]. AMPARs are composed of subunit pairs termed AC ("pore-proximal") and BD ("pore-distal"), where conformational changes in BD subunits dominate channel gating[26,70]. Consequently, the assembly of pore subunits dictates the functional interactions with auxiliary proteins depending on their location within the receptor complex. For example, while TARP γ2 has been shown to engage with the KGK sites of the BD pair via the β4-TM2 loop, the longer β1-β2 loop of TARP γ8 and GSG1L has been shown to simultaneously access the LBDs of adjacent subunit pairs[15,23,71]. Until recently, the occupancy of AC and BD pairs in the heteromer structure was not firmly established, but it is now thought that in GluA1/A2 heteromers, A1 and A2 sit at AC and BD positions, respectively[26,29] (Fig. 8). Based on their location, GluA1 and GluA2 exhibit differences in conformational rearrangements at the level of the LBD during open-to-desensitized transitions and contribute asymmetrically to channel gating[71]. This state-dependent asymmetry produces gating-state-specific interactions with flexible TARP extracellular loops (shown for TARP γ8), and presumably, GSG1L.

Using a mutational strategy combined with electrophysiology, we investigated this notion with A1/A2 heteromers bound by TARP γ2 or GSG1L auxiliary subunits (Figs. 6 and 7). Mutation of KGK in the GluA1 subunit did not produce a loss-of-function effect on TARP-dependent gating. When the KGK site was mutated in the GluA2 subunit, there was a partial attenuation of TARP modulation on desensitization kinetics and the equilibrium response, which was further pronounced when all AMPAR subunits were mutated. These findings showcase that the functional interaction of TARPs with GluA2/BD is dominant for gating, but a synergy exists between the core subunits. A similar subunit coordination was observed in the context of GSG1L and recovery from desensitization. When the E651 site was mutated in GluA1/AC, GSG1L continued to slow the recovery time course. Only when the equivalent residue was mutated in GluA2/BD was there a partial loss of GSG1L's effect, which was further attenuated by disrupting functional interactions at all four pore-forming subunits. These findings identify GluA2 as being primed to dominate channel gating and, as such, provide a structural framework to understand how AMPAR heteromers are modulated by all claudin family members.

## Methods

### Animals for histology
The transgenic GSG1L knockout (KO) rat was among the mutants generated by transposon-based mutagenesis by Kent Hamra at UT Southwestern[35], and was purchased and retained in-house. GSG1L KO rats were maintained by crossing GSG1L heterozygous animals in-house. All animal procedures were approved by the Vanderbilt University Animal Care and Use Committee and were in agreement with the NIH and Vanderbilt University guidelines for the care and use of laboratory animals.

### Genotyping by PCR of transgenic rats for histology
Genotyping was done as previously described[19]. PCR amplification was conducted with the following primers: GSG1L left: acgttgtagtgaccccaagc; GSG1L right: tgcacgcatactacaatga; SFB2: tcatcaaggaaaccctggac.

### Staining of whole brains
P14, P24, P42, and P120 GSG1L KO rat brains and corresponding wild-type (WT) rat brains ($n = 2$) were anesthetized with sodium pento-barbital (Nembutal). The animals were then perfused with normal Rat Ringer solution for 2 min for complete perfusion, followed by 4% paraformaldehyde in 0.1 M phosphate buffer for 4 min. The brains

were then obtained and permeabilized with 0.01% sodium deoxycholate and 0.02% Triton X-100 in PBS buffer for 2 h. Following permeabilization, the tissue samples were incubated with X-gal staining solution containing 5 mM $K_3[Fe(CN)_6]$, 5 mM $K_4[Fe(CN)_6]$, 2 mM $MgCl_2$, 0.02% Triton X-100, and 0.1% X-gal in PBS buffer at 37 °C for 8 h in the dark. Stained brains were imaged (MULTIZOOM AZ100M, Nikon) the following day (1 day post-staining).

### Staining of fixed brain sections
P14, P21, P60, P180, and P240 GSG1L KO rat brains ($n = 2$) were anesthetized with sodium pentobarbital (Nembutal). Each animal was then perfused with normal Rat Ringer solution for 2 min for complete perfusion, followed by 4% paraformaldehyde in 0.1 M phosphate buffer for 4 min. Obtained brains were further fixed for 15 min. Fixed brains were subsequently sectioned to generate 300 µm-thick coronal or sagittal sections using a vibratome (Leica VT 1200). The sections were then permeabilized with 0.01% sodium deoxycholate and 0.02% Triton X-100 in PBS buffer for 2 h. Following permeabilization, 300 µm-thick sections were incubated with X-gal staining solution containing 5 mM $K_3[Fe(CN)_6]$, 5 mM $K_4[Fe(CN)_6]$, 2 mM $MgCl_2$, 0.02% Triton X-100, and 0.1% X-gal in PBS buffer at 37 °C for 8 h in the dark. Stained brains were imaged (MULTIZOOM AZ100M, Nikon) the following day (1 day post-staining) and then left for 1 week at 4 °C to enhance the lacZ staining due to residual X-gal in the tissue. It should be noted that efforts to generate reliable anti-GSG1L antibodies for immunohistochemistry (IHC) have so far failed.

### Affinity purifications (APs)
Animal procedures were performed in accordance with national and institutional guidelines in accordance with the German law for the welfare of animals and were approved by local authorities (Regierungspräsidium Freiburg X14/14H). Brains from three adult (P59) GSG1L WT and KO rats were dissected and shock-frozen in liquid nitrogen. The following procedures were performed essentially as described in[6]. The brains were cut in pieces and homogenized in 30 ml ice-cold H-buffer (10 mM Tris/HCl pH 7.5, 300 mM sucrose, 1.5 mM $MgCl_2$, 1 mM EGTA, 1 mM iodoacetamide and protease inhibitors) with Dounce homogenizer. The nuclear fraction was removed by centrifugation (4 min at 1000xg) and the respective supernatants subjected to ultracentrifugation (20 min at 200,000xg). The pellets were homogenized in 30 ml Lysis buffer (5 mM Tris/HCl pH 7.4 supplemented with protease inhibitors) and incubated on ice for 30 min. Additional ultracentrifugation (20 min at 200,000xg) separates soluble proteins from the membrane/membrane-attached proteins. The latter was resuspended in 20 mM Tris/HCl pH 7.4 and loaded on top of a two-layer density gradient (0.5 M /1.3 M sucrose in 10 mM Tris/HCl pH 7.4). After ultracentrifugation (45 min at 30,000 rpm, Sorvall Surespin 630), the membrane protein concentrations were determined by Bradford assay and adjusted to 10 mg/ml.

For each affinity purification (Fig. 2a, b and Supplementary Fig. 2), 1.5 mg of membrane proteins from WT or GSG1L KO rats were solubilized with 1.5 ml CL-47 (Logopharm) supplemented with protease inhibitors (Aprotinin, Leupeptin, Pepstatin A, PMSF). Homogenates were cleared by ultracentrifugation (10 min at 125,000xg) and solubilisates were incubated with 15 µg antibodies pre-coupled to protein A Dynabeads. The following affinity purified GSG1L antibodies were used for the interactome analysis: Ab#1, polyclonal, raised in rabbit against C-term of rat GSG1L (aa304-322), Ab#2, polyclonal, raised in rabbit against rat GSG1L (aa257-278), Ab#3, polyclonal, raised in rabbit against rat GSG1L (aa287-308). Antibodies were incubated for 2 h with solubilisates and subsequently briefly washed two times with 0.5 ml CL-47 buffer. Proteins were eluted with 10 µl of Lämmli buffer w/o DTT. Eluted proteins were shortly separated on SDS-PAGE and silver stained.

For two-step APs (Fig. 2c), 0.5 mg of WT membranes were solubilized in 0.5 ml CL-91[6]. Solubilisates were first incubated for 2 h with a

mixture of GSG1L antibodies (20 μg Ab#2, 25 μg Ab#3 and 5 μg Ab#4 (Proteintech, #17328-1-AP)) coupled to protein A Dynabeads to completely affinity isolate GSG1L. After this incubation period, the solubilisate was transferred to a mixture of coupled AMPAR antibodies (25 μg anti-GluA1, #AB1504 Millipore; 20 μg anti-GluA2, #75-002 NeuroMab; 5 μg anti-GluA2/3, #07-598 Millipore; 5 μg anti-GluA3 #182203 Synaptic Systems; 10 μg anti-GluA4 #AB1508 Millipore) for 2 h of incubation. In both cases, after incubation, the beads with antibodies were separated, briefly washed, and proteins eluted as described above. At each step of this experiment, 10 μl aliquots were taken and analysed by SDS-PAGE and Western blotting to control for the complete affinity isolation of AMPARs.

## Mass spectrometry (MS)

The eluted proteins of each affinity purification were separated on SDS-PAGE and proteins subsequently visualized by silver staining. The gel lanes were cut and split in two sections to reduce complexity. Proteins were in-gel digested using sequencing-grade modified trypsin (Promega GmbH, Walldorf, Germany) following the procedure described in[72]. Vacuum-dried peptides were dissolved in 20 μl of 0.5% (v/v) trifluoroacetic acid, loaded onto a trap column (C18 PepMap100, 5 μm particles, Thermo Fisher Scientific GmbH, Dreieich, Germany), separated by reversed-phase chromatography via a 10 cm C18 column (PicoTip™ Emitter, 75 μm, tip: 8 μm, New Objective, self-packed with ReproSil-Pur 120 ODS-3, 3 μm, Dr. Maisch HPLC; flow rate 300 nl/min) using an UltiMate 3000 RSLCnano HPLC system (Thermo Scientific), and eluted by an aqueous organic gradient (eluent "A": 0.5% acetic acid; eluent "B": 0.5% acetic acid in 80% acetonitrile). MS-analyses were executed on an Orbitrap Elite mass spectrometer with a Nanospray Flex Ion Source (both Thermo Scientific). Precursor signals (LC-MS) were acquired with a target value of 1,000,000 and a nominal resolution of 240,000 (FWHM) at m/z 400; scan range 370–1700 m/z. LC-MS/MS data were extracted using "msconvert.exe" (part of ProteoWizard; http://proteowizard.sourceforge.io/). Peak lists were searched against a UniProtKB/Swiss-Prot database (containing all rat, mouse, and human entries) with peptide mass tolerance ± 5 ppm; fragment mass tolerance ± 0.8 Da using Mascot 2.6.2 (Matrix Science, UK). One missed trypsin cleavage and common variable modifications including S/T/Y phosphorylation were accepted for peptide identification. Significance threshold was set to $p < 0.05$.

## Quantification of proteins

Proteins were quantitatively evaluated according to a procedure described in ref. [73]. Briefly, peptide signal intensities (peak volumes, PVs) were extracted from FT full scans and mass calibrated using MaxQuant v1.6.3 (http://www.maxquant.org). Peptide PV elution times were then aligned and assigned to peptides based on matching m/z and elution times (tolerances 2-3 ppm / ±1 min) as described previously[74]. All protein-specific peptide signal intensities in all runs were further processed to eliminate the influence of PV outliers, false assignments, and gaps by exploring the consistency of PV relations within proteins (i.e., protein-specific PV ratios between and within runs). Abundance$_{norm}$ spec values (as a measure of molecular abundance) were calculated as described in[74].

Specificity of co-purifications (Fig. 2a and Supplementary Fig. 2) was determined according to target-normalized abundance ratios (tnRs) of proteins affinity purified from WT versus control (GSG1L KO) (calculated as described in ref. [73]). Six distinct anti-GSG1L APs (3 from WT and 3 from GSG1L KO) provided 3 ratios. This information was inspected using the BELKI software suite (https://github.com/phys2/belki). tnR values were visualized by t-distributed stochastic neighbor embedding (t-SNE; Fig. 2a). The high consistency of all datasets was also shown as 2D plots of tnR values (Supplementary Fig. 2). A minimum tnR of 0.25 was used as the indicator for specific interaction. Abundances of all proteins specifically and consistently affinity

purified in anti-GSG1L APs were plotted as relative values to the abundance of GSG1L, which was set to 1 (Fig. 2b); data are mean of three experiments. Two-step APs (Fig. 2c) were used to calculate the proportion of GSG1L-containing AMPARs relative to the entirety of AMPARs in the rat brain. First, the protein abundance values in target-depleting GSG1L APs and the subsequent target-depleting GluA1-4 APs were determined for all GSG1L interactors. Relative amounts of proteins in GSG1L and GluA1-4 APs (protein abundance in AP/sum of protein abundances in (GSG1L + GluA1-4 APs)) were calculated.

## Plasmids for recombinant electrophysiology

All AMPAR pore-forming subunits correspond to the rat sequence in the pRK5 vector. For GluA2 homomers, the Q/R unedited flip (GluA2Q$_{flip}$) isoform was used as indicated. For GluA1 homomers, the flip isoform was also used (GluA1$_{flip}$). For the study of heteromers, the Q/R edited flip (i.e., GluA2R$_{flip}$) isoform of GluA2 was used. Residue numbering includes the signal peptide. Mutant receptors were generated using site-directed mutagenesis and all new constructs were screened by restriction digest prior to confirmation by sequencing. Auxiliary subunits and AMPARs were co-expressed at a 2:1 or 2:1:1 cDNA ratio for homomers and heteromers, respectively, except for constructs in which mouse TARP γ2 or mouse GSG1L were tethered to the AMPAR subunit in the pRK5 vector (for γ2, refs. [21,24]); fusion notation /γ2 or /GSG1L vs. co-expression notation +γ2 or +GSG1L. For GSG1L fusion constructs, the full-length coding sequence of GSG1L and a short 7 aa linker sequence (same as the γ2 fusion construct, i.e. ELGTRGS[75]) were synthesized by Bio Basic Inc. (Ontario, CA) and subcloned into the GluA1 and GluA2 vectors. Species and vectors for co-expressed auxiliary subunits are as follows: human TARP γ7 in pCMV6-XL4 (OriGene), mouse GSG1L in pReceiver-M02 (GeneCopoeia), and mouse CKAMP44 in pRK5 (from J. von Engelhardt). cDNA was co-transfected with a plasmid encoding enhanced green fluorescent protein (eGFP) to identify transfected cells.

## Cell culture and transfection

HEK293T/17 cells (ATCC, CRL-11268) were maintained at 37 °C under 5% CO$_2$ in Minimum Essential Medium with GlutaMAX (i.e., MEM GlutaMAX) supplemented with 10% fetal bovine serum (FBS). Cells were plated at low density ($1.6 \times 10^4$ cells/ml) on poly-D-lysine-coated 35-mm dishes, and transiently transfected 24-48 h post-plating using the calcium phosphate precipitation method. After 6-12 h, cells were washed twice with divalent PBS and maintained in fresh medium containing 30 μM DNQX or NBQX to minimize auxiliary protein-induced cytotoxicity.

## Patch-clamp electrophysiology recordings

All recordings were performed 24-48 h post-transfection on excised outside-out patches. External solution contained (in mM): 150 NaCl, 5 HEPES, 0.1 CaCl$_2$ and 0.1 MgCl$_2$, and 2% phenol red at pH 7.3–7.4. Internal solution contained (in mM): 115 NaCl, 10 NaF, 5 HEPES, 5 Na$_4$BAPTA, 0.5 CaCl$_2$, 1 MgCl$_2$, and 10 Na$_2$ATP at pH 7.3-7.4. For GluA1/A2(R) heteromer recordings, 30 μM spermine was included in the internal solution in place of Na$_2$ATP; data were only included when the I-V plot was linear, typical of GluA2(R)-containing AMPARs[24,76] (see Supplementary Fig. 4). The osmotic pressure of all solutions was adjusted to 295-300 mOsm with sucrose.

Recording pipettes were composed of borosilicate glass (3-6 MΩ, King Precision Glass, Inc.) coated with dental wax. Agonist solution (10 mM L-Glu or 1 mM KA) was rapidly applied using a piezo-stack driven perfusion system (Physik Instrumente) and solution exchange (<400 μs) was determined by measuring the liquid junction current at the end of each experiment. All recordings were performed using an Axopatch 200B amplifier (Molecular Devices, LLC). The holding potential during recordings was −60 mV or −100 mV. Current records were filtered at 5 kHz and sampled at 25 kHz. Series resistance (3-12

MΩ) was compensated for by 95%. All experiments were performed at room temperature. Data were acquired using pClamp9 software (Molecular Devices, LLC).

## Fitting analysis of electrophysiological data

Electrophysiological recordings were analyzed using Clampfit 10.5 (Molecular Devices, LLC). Current decay rates were fit using 1st or 2nd order exponential functions of the form $y = A_i*exp(-x/\tau_i)$. For decay rates requiring 2nd order exponential fits, time constants are presented using both the individual components and as weighted means. To measure recovery from desensitization, a two-pulse protocol was used in which agonist was applied at variable interpulse intervals, and the peak amplitude of the second (test) pulse was expressed as a fraction of the peak amplitude of the first (initial) pulse. The recovery time course was fit with mono- or bi-exponential functions, as indicated, to measure $\tau_{recovery}$ (described in[45]; see also Results and Supplementary Tables 2 and 4). In all figures, current traces are normalized unless otherwise indicated. Data were illustrated using Origin 2020 (OriginLab) and Adobe Illustrator.

## Statistical analysis of electrophysiological data

Statistical details can be found in the figure legends and Source Data file. Data are presented as mean ± SEM, where *n* values refer to the number of individual patches. Data points correspond to individual patches for each transfection condition. Transfections were performed at least two independent times. Data were assessed for normality using a Shapiro-Wilks test and appropriate parametric or nonparametric tests were conducted accordingly. For simple pairwise comparisons, a two-tailed unpaired Student's *t*-test or Mann-Whitney *U* test was performed as indicated. For comparisons between multiple groups (parametric), a one-way between-subject ANOVA was performed. Subsequent pairwise comparisons were made using Tukey's Honestly Significant Difference (HSD) test. For comparisons between multiple groups (nonparametric), a Kruskal-Wallis ANOVA was performed, followed by pairwise Mann–Whitney *U* tests with application of a Bonferroni-Holm correction. Significance level was set at 0.05 and is denoted as *$p \leq 0.05$, **$p \leq 0.01$, and ***$p \leq 0.001$. Exact *p*-values are provided in the Source Data file for both significant and nonsignificant results, except when $p < 0.001$. Statistical analysis was conducted using SPSS Statistics (IBM) and custom statistical software kindly provided by Joe Rochford (McGill University).

## Reporting summary

Further information on research design is available in the Nature Portfolio Reporting Summary linked to this article.

# Data availability

Data supporting the findings of this study are available upon request. The mass spectrometry proteomics data have been deposited to the ProteomeXchange Consortium via the PRIDE partner repository[77] with the dataset identifier PXD044621 [10.6019/PXD044621]. Referred protein structures have the following PDB accession codes: 5WEO, 5VHY, 1FTJ, 5KBU, 7RYZ. Source data for Figs. 2, 3, 4, 5, 6, and 7 and Supplementary Figs. 2, 3, 5, 6, 7, and 8 are provided with the paper as indicated. Source data are provided with this paper.

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

## Acknowledgements

We thank Jakob von Engelhardt for kindly providing the CKAMP44 cDNA. We also thank all Bowie lab members, past and present, for thoughtful discussion and suggestions, especially Marika Arsenault, Yuhao Yan and Xin-tong Wang for contributing some patch recordings. We especially thank Mina Moniri and Dr. Phil Biggin (molecular dynamics, Oxford University), Drs. Hana Antonicka and Eric Shoubridge (blue native PAGE analysis, McGill), and Drs. Ryan Alexander and Arjun Bhaskaran (native recordings in the somatosensory cortex, Bowie lab) for performing exploratory experiments that were ultimately not included in this study. We acknowledge the use of the Nikon AZ100M at the Vanderbilt CISR Core Facility. This work was supported by an operating grant from the Canadian Institutes of Health Research (CIHR) (to D.B.), National Institutes of Health (NIH) R01 grants HD061543 and MH123474 (to T.N.), and grants of the Deutsche Forschungsgemeinschaft (DFG, German Research Foundation) SFB 1381 (project-ID 403222702), FA 332/15-1 and 16-1 (to B.F.). A.M.P. was supported by Natural Sciences and Engineering Research Council of Canada (NSERC) CGS-M and PGS-D fellowships.

## Author contributions

A.M.P and D.B. conceived the project; A.M.P. performed electrophysiology experiments and related data analyses; J.S. performed experiments related to protein biochemistry and proteomic analyses; A.K. performed histology experiments; A.M.P., J.S., and A.K. made figures and evaluated data. T.N., B.F., and D.B. evaluated data and supervised the work. A.M.P. and D.B. wrote the manuscript with the support of all authors.

## Competing interests

The authors declare no competing interests.
