## [Peer Review File · Nature Communications]

GSG1L-containing AMPA receptor complexes are defined by their spatiotemporal expression, native interactome and allosteric sitesReviewers' Comments:

Reviewer #1:

Remarks to the Author:

In their comprehensive study, Perozzo et al. made significant advances concerning GSG1L, a claudin protein associated with AMPA receptors (AMPA receptors). They found that GSG1L's expression within the rodent brain is not uniform but rather, restricted to particular regions, cell types, and developmental stages. This finding differs from the expression pattern of TARPs, another set of claudin proteins, which are ubiquitously expressed across most brain regions and throughout development. Perozzo et al. conducted high-resolution proteomic analyses on native receptor complexes, revealing that GSG1L assembles into a unique configuration with AMPARs, forming a set of low-abundance receptor complexes characterized by a distinct subunit composition. This assembly appears to be different from the one formed by TARPs, suggesting a divergent role for GSG1L. Further, they demonstrated that the primary actions of GSG1L are not mediated through the KGK motif, an evolutionarily-conserved regulatory site previously associated with AMPAR modulation. Instead, they discovered a separate, evolutionarily-conserved allosteric site that GSG1L employs for its activity. This groundbreaking finding shifts the understanding of how GSG1L interacts with and modulates AMPARs. In addition, Perozzo et al. highlighted the importance of cooperation between the pore-forming and auxiliary subunits of AMPARs for effective channel gating. They discovered that the assemblies formed by AMPAR-TARP and AMPAR-GSG1L rely on this interplay, with the GluA2 subunit playing a dominant role in fine-tuning the channel gating mechanism. This implies a complex and synergistic relationship between these subunits, adding another layer to the intricacies of AMPAR regulation in the brain. The study provides a comprehensive characterization of the AMPAR-GSG1L complex, which has been previously understudied. They identified GSG1L as a unique regulator of AMPAR-mediated neurotransmission, and provide evidence to support that GSG1L binds to all GluA subunits and can constitute the inner core alone or together with TARP/CNIH subunits. Moreover, Perozzo et al. provide deep insights into the regulation of AMPAR desensitization, demonstrating how TARP $\gamma 2$ and GSG1L modify this process. This information is critical for understanding the AMPAR-GSG1L signaling complex's specialized roles at select glutamatergic synapses. Their work underscores the need for future studies on the AMPAR-GSG1L complex's impact on AMPAR response fidelity, excitability, and circuit behavior.

There are only minor clarifications that would strengthen the study.

1. The use of lacZ as a reporter gene to estimate GSG1L expression might not reflect the true dynamics of the protein's expression. The lacZ signal only provides a proxy for GSG1L expression and might not accurately represent GSG1L's native expression patterns and levels. It's also possible that not all cells expressing GSG1L activate the reporter gene, leading to underestimations of GSG1L expression. The authors should consider this caveat in their results and discussion.

1) The authors have inferred the composition of AMPAR-TARP and AMPAR-GSG1L from proteomics data. However, this can be misleading because of affinity bias. Ideally, the authors could present an alternative method to discuss composition, or at the very least, acknowledge the issue.

2) Could the authors comment on the lack of native scaffolding proteins, post-translational modifications, and specific cellular environments in HEK cells that may affect the properties of the receptors and their modulation by TARPs and GSG1L.

3) Perozzo et al used tethering to manipulate AMPAR-TARP complexes. However, it is possible that this artificial manipulation could influence the behavior of the proteins in a way that is not representative of their natural state. For example, it might affect the spatial configuration or the dynamics of the protein interactions. Could the authors comment on the use of the AMPAR-TARP tethered complexes?

4) The authors state that there may be "some functional contribution by GluA1 masked by the dominance of GluA2," but this possibility isn't fully investigated. Although they acknowledge this limitation, they don't seem to account for it in their conclusions. This should be further discussed.

Reviewer #2:

Remarks to the Author:

Review of Perozzo et al.,

The manuscript by Perozzo et al., undertakes a comprehensive examination of GSG1L's tissue and temporal distribution, binding partners and biophysical properties. This work is an important advance in our understanding of an understudied and interesting class of auxiliary proteins, which diverge in curious ways from the well-characterized TARP family. The conclusions are strong, and well supported by high quality data and well narrated by nicely written text and figures. Overall, this is a strong paper with broad appeal for a general cellular neuroscience audience. The following comments aimed at improving the manuscript.

Major

The union of native analysis of GSG1L distribution and binding partners along with the recombinant functional characterization is powerful. However, the recombinant characterization should include more native-like protocols. Specifically, the authors should examine if the GSG1L also enables superactivation/resensitization during trains of stimuli (see Carbone and Plested, Figure 7, Nat Comms 2016) and if 3D and/or E658K block such an effect in g2 or GSG1L. Determining if GSG1L also supports superactivation (which has not been examined to my knowledge) and gaining some insight into the molecular basis of it, would be of tremendous value to the field, significantly broaden the impact of the work and our understanding of how TARPs and GSG1L might serve distinct physiological ends.

"Cooperativity" is used incorrectly, most notably line 346 but several other places. "Cooperativity" refers to when two things produce a bigger (or smaller) effect when applied together versus separately. However, this paper makes no mention of any sort of calculation to determine if the effect of one manipulation is greater (or lesser) in the presence of another than one would expect by simple addition. Some other term, one without the mechanistic implications, is needed.

The use of a Gal reporting rat is unexpected, particularly since several GSG1L antibodies and KO mouse are used in the same paper. Why was gal staining used and not just IHC? Are the anti-bodies not suitable for immuno? Some comment on this decision is appropriate.

Minor

The heteromeric experiments appear to have been done using GluA2(R) as judged by the methods section. That should be mentioned in the main Results text too.

Line 77, "modulation of AMPAR gating has been firmly tied to KGK motif". There are several aspects of modulation such as superactivation/resensitization, recovery, concentration-response curves, single channel conductance, open probability, KA efficacy which have not been firmly tied to this motif at this point. Better to change to "A number of aspects of modulation...". Same concern on Line 238.

Line 299, the authors mention that GSG1L continues to slow kinetics of E658K, presumably referring to both desensitization and deactivation. I suggest they explicitly mention both and not leave to the reader to go to the Table. This is important given deactivation is a component of the recovery process (Robert et al., 2005 J Neuro). Since E658K ablates the GSG1L slowing of recovery but does not change deactivation itself, the mutation effect appears specific to the exiting the desensitized state step not just the whole process of recovery. This is worth highlighting.

Line 99, it's unclear what is meant by this sentence. The use of 'cooperativity' here is leads the biophysically minded reader down a particular road for which there is no evidence.

Line 164, given the apparent increase in GSG1L over development, authors should mention in the Results what age the rats are when used for meAP-MS. Related, the authors comment on LRRT4 being a trans-synaptic scaffold but what about PRRT1? Nothing is mentioned about this high prevalence

binding partner.

Line 224, authors mention the slow component increased 3 or 14 fold when g7 or GSG1L are present. Is this effect statistically significant?

Line 388, the phrase "main actions". Why are some GSG1L modifications considered main and others not? Better to say "certain actions" or similar.

Line 459, authors say "the first extracellular loop of GSG1L, which is comprised of four α -helical bundles with an extracellular domain that includes a five-stranded β -sheet." There is likely a typo or edit error here since the first EC loop does not contain all that.

Line 989, vectors containing GluA1, 2 and tandems should be given.

Reviewer #3:
Remarks to the Author:
Perozzo et al., 2023

This manuscript by Perozzo et al presents interesting new details on the expression profile, interactome and functional properties of GSG1L-containing AMPA receptors. The study shows that GSG1L is included in approximately 5% of all AMPAR receptors in adult rats, but is generally expressed late in development and with regional specificity. Proteomic analyses revealed a seemingly distinct subunit composition and suggested that GSG1L acts as a single helper subunit in a significant proportion of native receptors. The authors also show that a new site on the AMPAR ligand binding region (E658K) is critical for the GSG1L-mediated slowing of desensitization kinetics. This collaborative study between multiple labs represents an iterative extension of their previous work – bringing together previous studies to present a more comprehensive, interdisciplinary account that offers new insights. Overall this study looks well carried out and Results mostly clearly presented. However, there are currently loose ends - some more serious than others. I think some further experiments and analysis are needed to tighten the data (as indicated), and revisions to the ms are required.

Specific points:

1. There are numerous examples of proteins in the CNS with characteristic distributions and properties. So, it is unclear what the title is conveying that is 'unique' to GSG1L.
2. Abstract, final sentence: the statement "these distinctions help explain GSG1L's evolutionary past" – is too much like window dressing, given the comment seems unsupported elsewhere.
3. Although lacZ/promoter activity can serve as a useful proxy for expression, it does not always accurately reflect protein levels. This caveat should be made clear in the presentation.
4. Line 141 – endopiriform cortex not labelled in Fig 1 or S1.
5. The consistent reference to 'cell-type specific' expression of GSG1L is unconvincing and indeed unsupported. While expression in the cerebellar granule cell layer is suggestive of cell-type specificity, this layer certainly does not contain only granule cells.
6. Line 161 – age of rats for proteomics? This is in the Methods but could be stated here.
7. Methods: the authors need to define the species of AMPAR and TARP plasmids used for functional

experiments.

8. the age of 'adult' rats used for affinity purifications needs to be given. This is critical as previous lacZ work has shown evidence of changes in expression during development. The authors should consider and discuss the fact that, from whole brain studies, this interactome may look very different at different time points/ages.

9. Some of the statements (lines 176-197) are a little abrupt and require more explanation for clarity. For example:

Line 176 – Although labelled as 'noteworthy' the significance of this statement is not elaborated.

Line 181 – Estimation of the stoichiometry needs to be more explicit/expansive.

Line 182 – The order could be reversed – GSG1L APs were used, not GluA2, so it seems more logical that the sentence should start "GluA2 appeared in an almost 1:1 ratio with GSG1L..."

10. Line 218 – The sentence appears poorly constructed, is not supported by the data to which it refers and is ambiguous. The actions of $\gamma 7$ and GSG1L are not 'qualitatively similar' to those of $\gamma 2$. This is hopefully not what you intended to imply, but neither are the actions of $\gamma 7$ and GSG1L 'qualitatively similar'. The problem is that the 'interpretation' at the start of the sentence relies heavily on data that is not shown (Table S1) and which is mentioned only in the subsequent sentence. This needs some re-structuring to avoid potential confusion.

11. Paragraph starting ln. 232: From the literature, GSG1L is not generally thought to attenuate equilibrium desensitization.

12. Figure 4F: The 3D mutation causes a speeding of desensitization in the absence of TARPs. This is important and requires a mention in the corresponding Results section (unless I have overlooked it, currently this seems to be lacking?).

13. Line 261 – the term "...shared..." is odd. It would be more accurate to say that "Together, these data demonstrate that the evolutionarily-conserved KGK motif is an allosteric site that is common target of claudin-related proteins ($\gamma 2$, $\gamma 7$, GSG1L), but not CKAMP44."

14. Ln. 294 – some context or comment is needed here to explain the reason why E658 was selected.

15. Regarding A1/2 heteromer experiments – To my mind it is crucial to measure and construct I/V relationship here to confirm the incorporation of both subunits. It is essential to have clear cut evidence that data were collected from a homogenous receptor population. If the authors already have such data, they should comment accordingly. If not the authors need to address this with more data.

16. Figure 6C: (see point 8 above). Is this effect independent of $\gamma 2$?

17. Figure 7: As mentioned above, evidence is really needed to exclude the possibility that experiments are on a heterogenous population of A1/GSG and A2/GSG receptors.

18. Figure 7B1 (inset on graph): I think the should perhaps read 'A1EK + A2'?

19. Figures C1, D1: As far as I can see the Graphs are missing legends for the control data?

20. Statistical tests: There are some issues with the presentation of statistical tests. ANOVA tests are mentioned but the results of these omnibus tests are not presented and in subsequent pairwise tests p-values are presented only as inequalities. This is not in line with the requirements of the Nature

portfolio Reporting Summary.

21. There is inappropriate data re-use and re-analysis in Figures 3, 4 and S3. The traces shown in Fig 4D are duplicates of those shown in Fig 3C. This is not explicitly stated and is only indirectly hinted at in the phrase in the legend of Fig 4D "...compared to the WT control...". Likewise, the wild-type data presented in Fig 4E and F is a duplicate of that in Fig 3D and E, this time compared to the 3D mutants. Again this is not explicitly stated. This duplication extends to Fig S3, where the wild-type data are re-used in another comparison, this time with E658K mutants. In effect, for each measure (SS/Pk, Tau des fast and % fast) there are five conditions for auxiliary subunits (none, $\gamma 2$, $\gamma 7$, GSG1L and CKAMP44) and three conditions for mutation (WT, 3D and EK). Data are replicated and analysed independently when they should be analysed en masse with a single set of Bonferroni-Holm corrections.

We would first like to thank all three reviewers for their thorough and thoughtful comments, which have greatly helped to improve our manuscript. We have carefully read their criticisms and revised our manuscript accordingly by making changes to the text and figures therein. In particular, we have addressed concerns related to the clarity of the language and interpretation of certain results. Through these revisions, we feel that we have fully addressed the queries and issues raised by each reviewer to meet the criteria required for consideration of our re-submission to *Nature Communications*.

A point-by-point rebuttal to the reviewers' comments follows below:

REVIEWER COMMENTS

Reviewer #1 (Remarks to the Author):

In their comprehensive study, Perozzo et al. made significant advances concerning GSG1L, a claudin protein associated with AMPA receptors (AMPA receptors). They found that GSG1L's expression within the rodent brain is not uniform but rather, restricted to particular regions, cell types, and developmental stages. This finding differs from the expression pattern of TARPs, another set of claudin proteins, which are ubiquitously expressed across most brain regions and throughout development. Perozzo et al. conducted high-resolution proteomic analyses on native receptor complexes, revealing that GSG1L assembles into a unique configuration with AMPARs, forming a set of low-abundance receptor complexes characterized by a distinct subunit composition. This assembly appears to be different from the one formed by TARPs, suggesting a divergent role for GSG1L. Further, they demonstrated that the primary actions of GSG1L are not mediated through the KGK motif, an evolutionarily-conserved regulatory site previously associated with AMPAR modulation. Instead, they discovered a separate, evolutionarily-conserved allosteric site that GSG1L employs for its activity. This groundbreaking finding shifts the understanding of how GSG1L interacts with and modulates AMPARs. In addition, Perozzo et al. highlighted the importance of cooperation between the pore-forming and auxiliary subunits of AMPARs for effective channel gating. They discovered that the assemblies formed by AMPAR-TARP and AMPAR-GSG1L rely on this interplay, with the GluA2 subunit playing a dominant role in fine-tuning the channel gating mechanism. This implies a complex and synergistic relationship between these subunits, adding another layer to the intricacies of AMPAR regulation in the brain. The study provides a comprehensive characterization of the AMPAR-GSG1L complex, which has been previously understudied. They identified GSG1L as a unique regulator of AMPAR-mediated neurotransmission, and provide evidence to support that GSG1L binds to all GluA subunits and can constitute the inner core alone or together with TARP/CNIH subunits. Moreover, Perozzo et al. provide deep insights into the regulation of AMPAR desensitization, demonstrating how TARP $\gamma 2$ and GSG1L modify this process. This information is critical for understanding the AMPAR-GSG1L signaling complex's specialized roles at select glutamatergic synapses. Their work underscores the need for future studies on the AMPAR-GSG1L complex's impact on AMPAR response fidelity, excitability, and circuit behavior.

Reply: We thank Reviewer #1 for their very positive perspective of our work.

There are only minor clarifications that would strengthen the study.

1. The use of lacZ as a reporter gene to estimate GSG1L expression might not reflect the true dynamics of the protein's expression. The lacZ signal only provides a proxy for GSG1L expression and might not accurately represent GSG1L's native expression patterns and levels. It's also possible that not all cells expressing GSG1L activate the reporter gene, leading to underestimations of GSG1L expression. The authors should consider this caveat in their results and discussion.

Reply: The point raised by the Reviewer is indeed correct. To alleviate the Reviewer's concern, we have added a few sentences to the revised Discussion to highlight this potential caveat (lines 425-430).

1) The authors have inferred the composition of AMPAR-TARP and AMPAR-GSG1L from proteomics data. However, this can be misleading because of affinity bias. Ideally, the authors could present an alternative method to discuss composition, or at the very least, acknowledge the issue.

Reply: We have estimated the 'average' composition of native GSG1L-containing AMPARs based on MS-quantification of subunits of affinity-isolated receptor complexes. In order to minimize potential biases from affinity purification, we applied three distinct anti-GSG1L antibodies with distinct epitopes across the primary sequence; this procedure effectively counteracts biases resulting from antibody-mediated interferences of protein-protein interactions that may finally lead to displacement/removal of auxiliary subunits. Nonetheless, we added a note of caution to the Discussion (lines 432-434). Moreover, we should

add that we are not aware of any other method superior to our MS-based approach for the analysis of the AMPAR composition in its native configuration(s).

2) Could the authors comment on the lack of native scaffolding proteins, post-translational modifications, and specific cellular environments in HEK cells that may affect the properties of the receptors and their modulation by TARPs and GSG1L.

Reply: It has been recognized for some time that HEK 293 cells provide a very good surrogate environment for the study of receptors and ion channels. The value of HEK 293 cells reflects the fact that the expression of exogenous proteins, e.g., AMPAR-auxiliary complexes, does not require additional factors for their proper trafficking or membrane insertion (discussed in Schwenk and Fakler, 2021 *J Physiol.*). Our lab and others have demonstrated that receptor responses from HEK 293 cells can faithfully recapitulate the properties of native AMPAR-TARP complexes, despite the absence of scaffolding proteins etc. that are found at central synapses (see next point below). In this context, less is known regarding GSG1L, though GSG1L was sufficient to induce short-term depression in a reconstituted system, consistent with enhanced short-term facilitation observed in thalamic neurons from GSG1L KO mice (Kamalova et al., 2020 *Cell Rep.*). We do acknowledge this point by the Reviewer regarding the added complexity that exists in the native system (e.g., the effect of lipids, transsynaptic scaffolding proteins, kinases etc.); however, within the context of the experiments performed in our studies, the data suggest that there is good agreement between the properties of recombinant receptors and our current understanding of native receptors.

3) Perrozo et al used tethering to manipulate AMPAR-TARP complexes. However, it is possible that this artificial manipulation could influence the behavior of the proteins in a way that is not representative of their natural state. For example, it might affect the spatial configuration or the dynamics of the protein interactions. Could the authors comment on the use of the AMPAR-TARP tethered complexes?

Reply: Indeed, the Reviewer raises a valid concern. However, co-expressed AMPAR and TARP subunits exhibit gating parameters that are consistent with tethered complexes saturated by TARPs (presented in this work in **Fig. 6**, see lines 340-341, and in our previous studies i.e., Dawe et al., 2016 *Neuron*). Moreover, our lab was able to recapitulate native AMPAR phenotypes observed in cerebellar Purkinje and stellate cells using tethered AMPAR-TARP complexes (Dawe et al., 2019 *Neuron*) consistent with other studies in the hippocampus (e.g., Shi et al., 2009 *Neuron*; Zhang et al., 2021 *Nature*). Therefore, within the context of our measurements, tethering of TARPs does not impact their assembly or function.

4) The authors state that there may be "some functional contribution by GluA1 masked by the dominance of GluA2," but this possibility isn't fully investigated. Although they acknowledge this limitation, they don't seem to account for it in their conclusions. This should be further discussed.

Reply: Our data show that for GluA1/A2 heteromer gating, the GluA2 subunit takes a more dominant role over the GluA1 subunit. We made this conclusion by investigating the impact of systematically mutating the regulatory sites for TARP $\gamma 2$ (i.e., the KGK site) or GSG1L (i.e., the EK site) on the GluA1 and GluA2 subunits (**Figs. 6 and 7**). From this work, we observed that mutating the KGK or EK sites on the GluA2 subunit had a greater effect whether looking at TARP or GSG1L modulation. In fact, when we performed the same mutations on the GluA1 subunit only, there was little-to-no effect. However, the near-complete loss of modulation by TARPs or GSG1L requires mutation of sites on *both* GluA1 and GluA2, accounting for our statement that "some functional contribution by GluA1 (is) masked by the dominance of GluA2". These points are elaborated in the Discussion where we also speculate on the structural basis for the dominance of GluA2 (lines 504-534).

Reviewer #2 (Remarks to the Author):

Review of Perozzo et al.,

The manuscript by Perozzo et al., undertakes a comprehensive examination of GSG1L's tissue and temporal distribution, binding partners and biophysical properties. This work is an important advance in our understanding of an understudied and interesting class of auxiliary proteins, which diverge in curious ways from the well-characterized TARP family. The conclusions are strong, and well supported by high quality data and well narrated by nicely written text and figures. Overall, this is a strong paper with broad appeal for a general cellular neuroscience audience. The following comments aimed at improving the manuscript.

We thank Reviewer #2 for their positive reception to the manuscript and suggestions for improvement.

Major

The union of native analysis of GSG1L distribution and binding partners along with the recombinant functional characterization is powerful. However, the recombinant characterization should include more native-like protocols. Specifically, the authors should examine if the GSG1L also enables superactivation/resensitization during trains of stimuli (see Carbone and Plested, Figure 7, Nat Comms 2016) and if 3D and/or E658K block such an effect in g2 or GSG1L. Determining if GSG1L also supports superactivation (which has not been examined to my knowledge) and gaining some insight into the molecular basis of it, would be of tremendous value to the field, significantly broaden the impact of the work and our understanding of how TARPs and GSG1L might serve distinct physiological ends.

Reply: We thank the Reviewer for proposing this interesting experiment; however, we feel that it lands outside the immediate scope of the functional work in the present study. Since we plan to perform a follow-up investigation that will focus on probing the role of GSG1L in a native context (e.g., in cerebellar GCs), this later study would be a more appropriate venue for the suggested experiments. As well, the proposed mechanism underlying TARP γ 8-mediated superactivation/resensitization is that TARPs promote the transition to high conductance and high open probability open states of the AMPAR, which is inconsistent with the reported suppressive effects of GSG1L on channel conductance and gating (McGee et al., 2015 *J Neurosci.*). Given this, we would not expect GSG1L-bound AMPARs to exhibit this property.

“Cooperativity” is used incorrectly, most notably line 346 but several other places. “Cooperativity” refers to when two things produce a bigger (or smaller) effect when applied together versus separately. However, this paper makes no mention of any sort of calculation to determine if the effect of one manipulation is greater (or lesser) in the presence of another than one would expect by simple addition. Some other term, one without the mechanistic implications, is needed.

Reply: We chose this term because we wanted to emphasize that the observed effect of mutating *both* GluA1 and GluA2 on TARP/GSG1L modulation cannot simply be explained by the additive phenotype of mutating GluA1 and GluA2 individually. However, we understand that the term ‘cooperativity’ may have biophysical and/or mechanistic implications that we do not directly study here, as the Reviewer rightly states. Therefore, we have modified our language using ‘subunit coordination’ or ‘subunit asymmetry’ throughout the manuscript text.

The use of a Gal reporting rat is unexpected, particularly since several GSG1L antibodies and KO mouse are used in the same paper. Why was gal staining used and not just IHC? Are the anti-bodies not suitable for immuno? Some comment on this decision is appropriate.

Reply: The Nakagawa lab has generated multiple polyclonal antibodies against GSG1L (Shanks et al., 2012 *Cell Rep.*). The batch of anti-GSG1L antibody used in the referenced publication (affinity purified from a particular batch of serum to stain the CA3 region of the adult rat) was completely depleted and no longer available in the lab. We therefore tried to do IHC using affinity purified antibodies obtained from the same epitope but from different animals. However, subsequent efforts to use various anti-GSG1L antibodies to conduct IHC have failed. As a result, we turned to the lacZ reporter.

In the Nakagawa lab, some anti-GSG1L antibodies were effective in conducting immunoprecipitation, but not optimal for IHC (Shanks et al., 2012 *Cell Rep.*; Kamalova et al., 2020 *Cell Rep.*). It is therefore not surprising that the Fakler lab was able to conduct mass spec (MS) experiments, albeit using antibodies from other sources. To clarify for the reader, we have inserted a sentence in the Methods (section ‘Staining

of fixed brain sections') to share the information that "efforts to generate reliable anti-GSG1L antibodies for IHC have so far failed".

Minor

The heteromeric experiments appear to have been done using GluA2(R) as judged by the methods section. That should be mentioned in the main Results text too.

Reply: Noted and amended (lines 329 and 335).

Line 77, "modulation of AMPAR gating has been firmly tied to KGK motif". There are several aspects of modulation such as superactivation/resensitization, recovery, concentration-response curves, single channel conductance, open probability, KA efficacy which have not been firmly tied to this motif at this point. Better to change to "A number of aspects of modulation...".. Same concern on Line 238.

Reply: The Reviewer is correct in that TARP modulation via the KGK motif is actually quite selective to certain aspects of gating, but not other receptor properties such as some of those mentioned and polyamine block (Dawe et al., 2016 *Neuron*). We have modified the language as suggested (lines 75 and 244).

Line 299, the authors mention that GSG1L continues to slow kinetics of E658K, presumably referring to both desensitization and deactivation. I suggest they explicitly mention both and not leave to the reader to go to the Table. This is important given deactivation is a component of the recovery process (Robert et al., 2005 *J Neuro*). Since E658K ablates the GSG1L slowing of recovery but does not change deactivation itself, the mutation effect appears specific to the exiting the desensitized state step not just the whole process of recovery. This is worth highlighting.

Reply: We appreciate the Reviewer's suggestion. To emphasize that the mutation of E658 to K selectively affects GSG1L-induced slowing of recovery from desensitization, but not entry into desensitization or deactivation kinetics, we added two panels to **Supplementary Fig. 3** (see **d-e**). This specificity is also mentioned in lines 314 and 323-324.

Line 99, it's unclear what is meant by this sentence. The use of 'cooperativity' here leads the biophysically minded reader down a particular road for which there is no evidence.

Reply: We have modified our terminology. Please see our response to the similar point above under 'Major' heading (second comment).

Line 164, given the apparent increase in GSG1L over development, authors should mention in the Results what age the rats are when used for mAP-MS. Related, the authors comment on LRRT4 being a trans-synaptic scaffold but what about PRRT1? Nothing is mentioned about this high prevalence binding partner.

Reply: We agree and have added the information on the age of rats used (P59) to the Methods.

PRRT1 was not further discussed as this protein interacts with/binds to a variety of molecularly-distinct AMPA receptor complexes and does not display any obvious preference for GSG1L, in contrast to LRRT4 (**Fig. 2c**).

Line 224, authors mention the slow component increased 3 or 14 fold when g7 or GSG1L are present. Is this effect statistically significant?

Reply: We did not formally conduct the statistics on the percent contributions of the current decay components. In some cases for GluA2 receptors alone, the contribution of the slow component was very small (0-1%) and thus it was more intuitive to describe the change in proportion by fold changes, as we have published previously (Perozzo et al., 2023 *J Neurosci*). Nonetheless, the increase in the slow component of GluA2 desensitization by $\gamma 7$ ($U_{(20,11)} = 202$, $p < 0.001$) and GSG1L ($U_{(20,15)} = 295$, $p < 0.001$) is statistically significant in each case (Bonferroni-Holm corrected p -values).

Line 388, the phrase "main actions". Why are some GSG1L modifications considered main and others not? Better to say "certain actions" or similar.

Reply: Noted and amended to be more specific (line 408).

Line 459, authors say “the first extracellular loop of GSG1L, which is comprised of four α -helical bundles with an extracellular domain that includes a five-stranded B-sheet.” There is likely a typo or edit error here since the first EC loop does not contain all that.

Reply: We thank the Reviewer for making us aware of this typo/edit error. We have now correctly described the structure of the first extracellular loop (line 484).

Line 989, vectors containing GluA1, 2 and tandems should be given.

Reply: We have now provided the plasmid vectors (pRK5) in the Methods.

Reviewer #3 (Remarks to the Author):

Perozzo et al., 2023

This manuscript by Perozzo et al presents interesting new details on the expression profile, interactome and functional properties of GSG1L-containing AMPA receptors. The study shows that GSG1L is included in approximately 5% of all AMPAR receptors in adult rats, but is generally expressed late in development and with regional specificity. Proteomic analyses revealed a seemingly distinct subunit composition and suggested that GSG1L acts as a single helper subunit in a significant proportion of native receptors. The authors also show that a new site on the AMPAR ligand binding region (E658K) is critical for the GSG1L-mediated slowing of desensitization kinetics. This collaborative study between multiple labs represents an iterative extension of their previous work – bringing together previous studies to present a more comprehensive, interdisciplinary account that offers new insights. Overall this study looks well carried out and Results mostly clearly presented. However, there are currently loose ends - some more serious than others. I think some further experiments and analysis are needed to tighten the data (as indicated), and revisions to the ms are required.

We thank Reviewer #3 for their complimentary view of our manuscript and helpful critique.

Specific points:

1. There are numerous examples of proteins in the CNS with characteristic distributions and properties. So, it is unclear what the title is conveying that is 'unique' to GSG1L.

Reply: This is a fair point; we have changed the title accordingly.

2. Abstract, final sentence: the statement "these distinctions help explain GSG1L's evolutionary past" – is too much like window dressing, given the comment seems unsupported elsewhere.

Reply: We have added a statement to the Introduction (lines 72-73) to support this reference to evolution in the Abstract. We included this statement in the Abstract since we feel that this is an interesting and worthy point of consideration in light of TARPs and GSG1L being evolutionarily-related and the findings presented in our manuscript. We also refer to this point in the Discussion (lines 481-483).

3. Although lacZ/promoter activity can serve as a useful proxy for expression, it does not always accurately reflect protein levels. This caveat should be made clear in the presentation.

Reply: We fully agree, and the caveat is now made clear in the revised manuscript (see Discussion lines 425-430). Please also refer to our response to the first comment from Reviewer #1 above.

4. Line 141 – endopiriform cortex not labelled in Fig 1 or S1.

Reply: We regret that our figure did not have the appropriate label. While revising the manuscript, we noticed that the correct anatomical name is endopiriform nucleus (EPN), not endopiriform cortex. We have both made the correction and added a label in **Fig. 1b**.

5. The consistent reference to 'cell-type specific' expression of GSG1L is unconvincing and indeed unsupported. While expression in the cerebellar granule cell layer is suggestive of cell-type specificity, this layer certainly does not contain only granule cells.

Reply: We agree and will refer to the expression pattern as region-specific but not cell-type specific. In the cerebellar granule layer (CGL), however, the most probable expression profile is granule cell (GC)-specific, as they are the most abundant cells. There are other cell types, such as unipolar brush cells in the CGL, but they are much less abundant. Nonetheless, we have revised the description accordingly throughout the manuscript text.

6. Line 161 – age of rats for proteomics? This is in the Methods but could be stated here.

Reply: We have added the information on the age of rats used (P59; see also point 8 below).

7. Methods: the authors need to define the species of AMPAR and TARP plasmids used for functional experiments.

Reply: We have now provided the species for all sequences in the Methods. Please also see the final comment related to plasmids from Reviewer #2 above.

8. the age of 'adult' rats used for affinity purifications needs to be given. This is critical as previous lacZ work has shown evidence of changes in expression during development. The authors should consider and discuss the fact that, from whole brain studies, this interactome may look very different at different time points/ages.

Reply: We have analyzed GSG1L complexes in rats at P59, which allowed for the analysis of the whole brain GSG1L interactome and should give, according to the lacZ staining, a fairly good representation of the average receptor composition.

The developmental increase in GSG1L expression may coincide with changes in its molecular environment. We may have missed proteins if: 1) they assemble with GSG1L at early timepoints only or 2) developmentally-dependent, cell-type-specific expression is linked to unique protein assemblies. Both are speculative, but we have added a comment in the Discussion section (lines 441-444).

9. Some of the statements (lines 176-197) are a little abrupt and require more explanation for clarity. For example:
Line 176 – Although labelled as 'noteworthy' the significance of this statement is not elaborated.
Line 181 – Estimation of the stoichiometry needs to be more explicit/expansive.
Line 182 – The order could be reversed – GSG1L APs were used, not GluA2, so it seems more logical that the sentence should start "GluA2 appeared in an almost 1:1 ratio with GSG1L..."

Reply: We agree that more explanation was needed here. We have now addressed all points and provided extended descriptions for more clarity in the Results section (lines 177 onwards).

10. Line 218 – The sentence appears poorly constructed, is not supported by the data to which it refers and is ambiguous. The actions of $\gamma 7$ and GSG1L are not 'qualitatively similar' to those of $\gamma 2$. This is hopefully not what you intended to imply, but neither are the actions of $\gamma 7$ and GSG1L 'qualitatively similar'. The problem is that the 'interpretation' at the start of the sentence relies heavily on data that is not shown (Table S1) and which is mentioned only in the subsequent sentence. This needs some re-structuring to avoid potential confusion.

Reply: We have removed the language of 'qualitatively similar', which we agree was unclear and potentially confusing, and instead refer specifically to entry into desensitization (line 223).

11. Paragraph starting Ln. 232: From the literature, GSG1L is not generally thought to attenuate equilibrium desensitization.

Reply: We have modified the paragraph as to not generalize to all claudin-related proteins (line 237).

12. Figure 4F: The 3D mutation causes a speeding of desensitization in the absence of TARPs. This is important and requires a mention in the corresponding Results section (unless I have overlooked it, currently this seems to be lacking?).

Reply: We did note this speeding in the present study as well as in the publication reporting the initial observation (Dawe et al., 2016 *Neuron*), but have now made this point explicit in the Results text (line 253).

13. Line 261 – the term "...shared..." is odd. It would be more accurate to say that "Together, these data demonstrate that the evolutionarily-conserved KGK motif is an allosteric site that is common target of claudin-related proteins ($\gamma 2$, $\gamma 7$, GSG1L), but not CKAMP44."

Reply: Agreed and amended.

14. Ln. 294 – some context or comment is needed here to explain the reason why E658 was selected.

Reply: We have added new text to the Results to clarify this point (lines 307-311). We screened a few charged residues in proximity to the KGK motif, where the ExL1 of GSG1L may be able to engage, using a mutagenesis strategy. The E658K mutant yielded interesting and interpretable results, which is why it was investigated further.

15. Regarding A1/2 heteromer experiments – To my mind it is crucial to measure and construct I/V relationship here to confirm the incorporation of both subunits. It is essential to have clear cut evidence that data were collected from a homogenous receptor population. If the authors already have such data, they should comment accordingly. If not the authors need to address this with more data.

Reply: We agree that demonstration of linear I-Vs is critical to confirm heteromerization/incorporation of the GluA2(R) subunit in AMPAR assemblies; for this reason, we included representative I-V traces and plots in the first iteration of the manuscript (please refer to **Supplementary Fig. 4**). To ensure this point is clear to the reader, we have also added a description to the Results text (lines 329-331).

16. Figure 6C: (see point 8 above). Is this effect independent of $\gamma 2$?

Reply: As point 8 refers to proteomics, we believe the Reviewer is referring to point 12 above. If so, for GluA1/A2 heteromers and GluA1 homomers, the desensitization kinetics of 3D mutants are modestly faster than WT (**Supplementary Table 3**). However, as for GluA2, this does not fully account for the attenuation of the TARP $\gamma 2$ effects on desensitization:

For A1/A2 heteromers, the 3D mutation on all subunits results in a 1.3-fold speeding of the fast component of decay compared to WT (3.6 vs. 4.6 ms), but in the presence of TARPs, current decay is 1.9-fold faster for 3D mutant receptors (4.1 vs. 7.7 ms). For GluA1 AMPARs alone, the 3D mutation results in a 1.5-fold speeding of the fast component of current decay (1.9 vs. 2.9 ms), but in the presence of TARPs, current decay is 2.8-fold faster for 3D vs. WT (2.6 vs. 7.3 ms).

Moreover, the equilibrium response is near-identical for WT and 3D receptors alone across all receptor combinations studied here, but large TARP-mediated steady-states are reduced by mutation of the KGK motif (**Fig. 4e** and **Supplementary Tables 1** and **3**). Bringing together structural data (Zhao et al., 2016 *Nature*; Twomey et al., 2016 *Science*) and our functional work, the KGK motif is a key molecular player in TARP modulation of AMPAR gating, in addition to residues in the TMD and TMD-LBD linkers.

17. Figure 7: As mentioned above, evidence is really needed to exclude the possibility that experiments are on a heterogenous population of A1/GSG and A2/GSG receptors.

Reply: As commented above (point 15), we assessed heteromerization by examining I-V relationships in the presence of intracellular spermine. Given that GSG1L enhances polyamine block of CP-AMPARs (McGee et al., 2015 *J Neurosci.*), as we show here for GluA1 homomers (**Supplementary Fig. 8g, h**), and slows recovery of GluA1 homomers by nearly an order of magnitude (time scale of seconds; **Supplementary Table 2**), it is unlikely that A1/GSG is contributing to these heteromer recordings. Given the low conductance/surface expression of GluA2(R) homomers (Greger et al., 2002 *Neuron*), it is also unlikely that GluA2(R)/GSG is a contributing factor.

18. Figure 7B1 (inset on graph): I think the should perhaps read 'A1EK + A2'?

Reply: The label is correct in that the grey data is A1 EK + A2 EK. Given that this combination of subunits represents the 'extreme' (i.e., all mutated) and the recovery time course was similar to that of A1 + A2, we did not study A1 EK + A2 or A1 + A2 EK receptors alone. We have modified the figure to make this clear.

19. Figures C1, D1: As far as I can see the Graphs are missing legends for the control data?

Reply: The labels have now been modified on the individual panels and are explained in the legend.

20. Statistical tests: There are some issues with the presentation of statistical tests. ANOVA tests are mentioned but the results of these omnibus tests are not presented and in subsequent pairwise tests p-values are presented only as inequalities. This is not in line with the requirements of the Nature portfolio Reporting Summary.

Reply: We thank the Reviewer for bringing this to our attention. We provided inequalities for initial submission, but always intended to provide this information for a revised/final submission. The pertinent statistical details are now included in the Source Data file and described in the Methods.

21. There is inappropriate data re-use and re-analysis in Figures 3, 4 and S3. The traces shown in Fig 4D are duplicates of those shown in Fig 3C. This is not explicitly stated and is only indirectly hinted at in the phrase in the legend of Fig 4D "...compared to the WT control...". Likewise, the wild-type data presented in Fig 4E and F is a duplicate of that in Fig 3D and E, this time compared to the 3D mutants. Again this is not explicitly stated. This duplication extends to Fig S3, where the wild-type data are re-used in another comparison, this time with E658K mutants. In effect, for each measure (SS/Pk, Tau des fast and % fast) there are five conditions for auxiliary subunits (none, γ 2, γ 7, GSG1L and CKAMP44) and three conditions for mutation (WT, 3D and EK). Data are replicated and analysed independently when they should be analysed en masse with a single set of Bonferroni-Holm corrections.

Reply: WT data shown in **Fig. 3** was used in **Fig. 4** and **Supplementary Fig. 3**, simply for reference/comparison purposes with the test conditions. We apologize if this was not made clear in our initial submission. We have now modified the figure legends to make this point clearer and avoid any confusion or misunderstanding.

In terms of statistical analysis, it did not make sense to perform analysis en masse since the 3D and EK mutations are effectively independent from one another. Accordingly, we chose to analyze and present the data separately, making comparisons back to the respective mutant receptor alone or in some cases to the WT receptor. These details are now fully described in the figure legends and Source Data file.

Reviewers' Comments:

Reviewer #1:

Remarks to the Author:

The authors have addressed all my concerns well.

Reviewer #2:

Remarks to the Author:

The authors have satisfied all my original concerns.

However, please take another look at line 223. Here the authors state that g7 and GSG1L slow entry to desensitization, just like g2 but to a lesser extent. They cite references 7, 46 and 47. Ref 7 is about GSG1L, 46 and 47 are about TARPs. However, there is no desensitization data for g7 in ref 46 that I can see. All kinetic data are for g5 in that paper. And ref 47 is a review. The review contains a table indicating g7 slows desensitization but the source of this original data is unclear. A better reference indicating that g7 slows desensitization is needed here.

Reviewer #3:

Remarks to the Author:

NCOMMS-23-21277A Perozzo et al.

I think the authors have generally done a good job on revising this ms. A couple of points remain. I feel the final paper would be improved by correcting these:

Point 17: This rationale is not totally convincing. While the supplementary IVs do demonstrate the possibility of getting a homogeneous population of heteromers, these are not the same plasmids as used in Figure 7. It still remains a possibility that a subset of GluA1/GSG containing receptors could be contributing to these slow kinetics. For clarity and accuracy the authors need to avoid seeming to gloss over this.

Point 21: The authors now explicitly comment in the figure legends on the re-use of traces. This is fine, but they need to be similarly explicit on the re-showing of data – i.e. state that the filled bars for A2 data in Figure 4e and f are repeated from Figure 3d and e.

Presentation aside, I remain unconvinced as to the validity of three separate statistical analyses for these data. Leaving aside the independence of the 3D (Fig 4) and EK (Fig S3) mutations, the 'control' data presented in Fig 3 is simply a subset of the data in Fig 4e and 4f. To my mind this should be analysed as one set of corrected pairwise comparisons (A2:3D plus A2:auxiliary comparisons for WT and 3D). If the authors wish to analyse the EK (Fig S3) data separately then they should provide detailed statistical justification in the Methods. Input from a statistician may be useful to resolve this point.

REVIEWERS' COMMENTS

Reviewer #1 (Remarks to the Author):

The authors have addressed all my concerns well.

Reply: We thank Reviewer #1 for their previous comments which helped to improve our manuscript.

Reviewer #2 (Remarks to the Author):

The authors have satisfied all my original concerns.

Reply: We thank Reviewer #2 for their previous comments which strengthened our manuscript.

However, please take another look at line 223. Here the authors state that g7 and GSG1L slow entry to desensitization, just like g2 but to a lesser extent. They cite references 7, 46 and 47. Ref 7 is about GSG1L, 46 and 47 are about TARPs. However, there is no desensitization data for g7 in ref 46 that I can see. All kinetic data are for g5 in that paper. And ref 47 is a review. The review contains a table indicating g7 slows desensitization but the source of this original data is unclear. A better reference indicating that g7 slows desensitization is needed here.

Reply: We have provided additional references as requested: Kato et al (2007) *J Neurosci.* (ref. Fig. 5) and Dawe et al (2016) *Neuron* (ref. Figs. 6 and 7).

Reviewer #3 (Remarks to the Author):

I think the authors have generally done a good job on revising this ms. A couple of points remain. I feel the final paper would be improved by correcting these:

Reply: We thank Reviewer #3 for their positive reception to our revised manuscript and further suggestions.

Point 17: This rationale is not totally convincing. While the supplementary IVs do demonstrate the possibility of getting a homogeneous population of heteromers, these are not the same plasmids as used in Figure 7. It still remains a possibility that a subset of GluA1/GSG containing receptors could be contributing to these slow kinetics. For clarity and accuracy the authors need to avoid seeming to gloss over this.

Reply: The corresponding I-V plots for the receptor combinations depicted in Fig. 7a₁-d₁ are shown below:

Given that GSG1L enhances polyamine block of GluA1 homomers (as shown in Supp. Fig. 8h), it is unlikely that A1/GSG1L receptors are significantly contributing to our patch recordings since the I-V relationships are linear in each case.

Point 21: The authors now explicitly comment in the figure legends on the re-use of traces. This is fine, but they need to be similarly explicit on the re-showing of data – i.e. state that the filled bars for A2 data in Figure 4e and f are repeated from Figure 3d and e.

Reply: We have added this information to the figure legend and it is also stated in the Source Data file.

Presentation aside, I remain unconvinced as to the validity of three separate statistical analyses for these data. Leaving aside the independence of the 3D (Fig 4) and EK (Fig S3) mutations, the 'control' data presented in Fig 3 is simply a subset of the data in Fig 4e and 4f. To my mind this should be analysed as one set of corrected pairwise comparisons (A2:3D plus A2:auxiliary comparisons for WT and 3D). If the authors wish to analyse the EK (Fig S3) data separately then they should provide detailed statistical justification in the Methods. Input from a statistician may be useful to resolve this point.

Reply: We thank Reviewer #3 for their suggestion on furthering the statistical analyses. We have now consulted with colleagues having expertise in math/statistics and they agreed that the statistical analyses presented in the manuscript were sufficient and appropriate. They agreed that our analysis shed light on comparisons between biologically-meaningful receptor combinations. They did not feel it was necessary to perform extensive pairwise comparisons as suggested by Reviewer #3. Given this advice, we did not perform further statistical comparisons. Details of the analyses have been provided in the Methods and Source Data file.